



# Measurement report: Rapid oxidation of phenolic compounds by O₃ and HO•: effects of air-water interface and mineral dust in tropospheric chemical processes

Yanru Huo[a], Mingxue Li[b], Xueyu Wang[c], Jianfei Sun[d], Yuxin Zhou[a],

Yuhui Ma[a], Maoxia He[a],*

[a] Environment Research Institute, Shandong University, Qingdao 266237,

P. R. China

[b] Department of Civil and Environmental Engineering, The Hong Kong

Polytechnic University, Hong Kong SAR, China

[c] College of Geography and Environmental Sciences, Zhejiang Normal

University, Jinhua 321004, China

[d] School of Environmental and Materials Engineering, Yantai University,

Yantai, 264005, PR China

_______________________________________________________

*Corresponding author: Prof. Maoxia He

Tel: 86-532-58631972 (o)

Fax: 86-532-5863 1986

E-mail address: hemaox@sdu.edu.cn





## Abstract

Environmental media affect the atmospheric oxidation processes of phenolic compounds (PhCs) released from biomass burning in the troposphere. Phenol (Ph), 4-hydroxybenzaldehyde (4-HBA), and vanillin (VL) are chosen as model compounds to investigate their reaction mechanism and kinetics at the air-water (A-W) interface, on $TiO_2$ clusters, in the gas phase, and in bulk water using a combination of molecular dynamics simulation and quantum chemical calculations. Of them, Ph was the most reactive one. The occurrence percentages of Ph, 4-HBA, and VL staying at the A-W interface are ~72%, ~68%, and ~73%, respectively. As the size of $(TiO_2)_n$ clusters increases, the adsorption capacity decreases until $n > 4$, and beyond this, the capacity remains stable. A-W interface and $TiO_2$ clusters facilitate Ph and VL reactions initiated by the $O_3$ and $HO^\bullet$, respectively. However, oxidation reactions of 4-HBA are little affected by environmental media because of its electron-withdrawing group. The $O_3$- and $HO\bullet$-initiated reaction rate constant ($k$) values follow the order of $Ph_{A-W} > VL_{TiO_2} > VL_{A-W} > 4\text{-}HBA_{A-W} > 4\text{-}HBA_{TiO_2} > Ph_{TiO_2}$ and $VL_{TiO_2} > Ph_{A-W} > VL_{A-W} > 4\text{-}HBA_{TiO_2} > Ph_{TiO_2} > 4\text{-}HBA_{A-W}$, respectively. Some byproducts are more harmful than their parent compounds, so should be given special attention. This work provides key evidence for the rapid oxidation observed in the $O_3/HO^\bullet$ + PhCs experiments at the A-W interface. More importantly, differences in oxidation of PhCs by different



environmental media due to the impact of substituent groups were also
identified.
**Keywords:** Air-water interface; Titanium dioxide (TiO$_2$); Phenolic
Compounds; Adsorption mechanism; Molecular dynamics (MD).
**1. Introduction**
Biomass burning, stemming from natural wildfires and human activity,
significantly contributes to atmospheric particulate matter (PM). Biomass
burning is a primary source of approximately 90% of the global primary
organic aerosols (POA) and releases a substantial quantity of organic
pollutants (Ito and Penner, 2005; Chen et al., 2017; Chen et al., 2023).
Biomass burning is to blame for about 62% of total annual emissions of
about 8.0 Tg of black carbon and 93% of total annual emission of about
33.9 Tg of organic carbon worldwide (Bond et al., 2004). Emissions from
biomass combustion are one of the primary sources of atmospheric and
particle pollutants that negatively affect human health, air quality, and
climate (Reid et al., 2005; Yao et al., 2016). One of the three main types of
biopolymers responsible for the formation of biomass is lignin (Sun et al.,
2011), also the polymeric organic molecule most abundant in plants (Lou
et al., 2010; Soongprasit et al., 2020). Pyrolysis of lignin releases phenolic
compounds (PhCs) into the air, including phenols, phenolic aldehydes, and
methoxyphenols. By mass, these PhCs make up between 21% and 45% of
the aerosol composition (Hawthorne et al., 1989; Diehl et al., 2013; Liao



et al., 2020; Soongprasit et al., 2020). Methoxyphenols are one of the
potential tracers that can be found in atmospheric wood smoke pollution,
with the emission rate ranging from 900 to 4200 mg kg$^{-1}$ fuel (Hawthorne
et al., 1989; Rogge et al., 1998; Simoneit, 2002; Chen et al., 2017).
Evidence shows that the oxidation processes of PhCs can result in the
formation of secondary organic aerosol (SOA) (Yee et al., 2013; Jiang et
al., 2023). Hence, it is imperative to explore the effects of PhCs when
exposed to atmospheric oxidants.
After being released into the atmosphere, PhCs will be oxidized by
ozone ($O_3$) and hydroxyl radicals (HO$^\bullet$). Both are significant contributors
to SOA. The homogenous oxidation of PhCs has been the emphasis of
previous studies (Henry et al., 2008; Yee et al., 2013; Liu et al., 2019;
Arciva et al., 2022). Researchers investigated the kinetics and reaction
mechanisms of gas-phase interactions of PhCs with $O_3$ and HO$^\bullet$ in the past
decade (Kroflič et al., 2018; Smith et al., 2016; Sun et al., 2021a; Sun et
al., 2021b; Liu et al., 2022). Furthermore, they investigated the
hydroxylation, ring opening, and oligomerization processes of PhCs in the
atmospheric liquid phase, with a focus on the potential environmental
toxicity and climatic effects of these events (Liu et al., 2022; Arciva et al.,
2022; Carena et al., 2023).
However, there is a dearth of specific data as well as explanations of
the mechanisms involved in the atmospheric oxidation of PhCs at the air-





water (A-W) interface. The atmosphere contains a high concentration of
aqueous aerosols and water microdroplets (Zhong et al., 2019). The
oxidation of PhCs can rapidly occur at A-W interface. The term "water
surface catalysis" denotes the phenomenon where chemical reactions
happen at a faster rate at A-W interface compared to the bulk phase (Lee
et al., 2015a; Lee et al., 2015b; Yan et al., 2016; Banerjee et al., 2017). In
chemical engineering, titanium dioxide ($TiO_2$) is an essential photoactive
component found in atmospheric mineral dust (Sakata et al., 2021; Wang
et al., 2023). The interaction between PhCs and $TiO_2$ is continuous
(Grassian, 2009; Rubasinghege et al., 2010; Shang et al., 2021), despite the
relatively low prevalence of $TiO_2$ mineral particles (comprising 0.1% to 10%
by mass). Therefore, it is essential to investigate the disparity in the
oxidation reaction mechanisms and kinetics of PhCs at A-W interface and
mineral dust particles.
Phenol (Ph), 4-hydroxybenzaldehyde (4-HBA), and vanillin (VL) are
selected as model compounds to present comprehensive mechanistic
information at A-W interface, on $TiO_2$ clusters, in the gas phase, and in
bulk water, using a combination of molecular dynamics simulation and
quantum chemical calculations. Rate constants were measured throughout
a wide temperature range in various EM. Additionally, computational
toxicology was employed to evaluate the ecotoxicological impact of PhCs
and their transformation products.





## 2. Methods

### 2.1 Molecular dynamics simulation

All of the molecular-dynamics simulations were carried out by utilizing the GROMACS 2019 package, which included the AMBER force field. Parametrization of the Ph, 4-HBA, and VL was accomplished by using the GAFF force field in conjunction with RESP charge calculations performed at the M06-2X/6-311++G(3df,2p)//M06-2X/6-31+G(d,p) level. The TIP3P water model was utilized so that individual water molecules may be represented (Jämbeck and Lyubartsev, 2014).

### 2.1.1 Properties of Ph, 4-HBA, and VL at the A-W interface

Considering the significance of the interfacial behavior of Ph, 4-HBA, and VL at the A-W interface, the properties of these three substances were initially examined by focusing on the A-W interface. **Fig. S1 (a)** depicts a rectangular box that has dimensions of $4 \times 4 \times 9$ nm$^3$ and has a Z-axis that is perpendicular to the A-W contact. This box was used for all simulations. To begin the process of constructing the initial configurations, a water slab measuring $4 \times 4 \times 4$ nm$^3$ was positioned at the coordinates (2 nm, 2 nm, 4.5 nm) of the center of mass (COM). Because the rest extension along the Z-axis of the box was sufficiently large (2.5 nm$^3$), it was possible to steer clear of the intersection of two A-W interface. Prior to the formal simulation, six Ph molecules were randomly placed in a vacuum above the



water box for 150 nanoseconds of NVT molecular-dynamics simulations.
The results show that there are no significant π-π interactions or formation
of hydrogen bonds between the Ph molecules. To simplify the model, this
was followed by simulations of individual molecules. Ph, 4-HBA, or VL
were each placed in their own compartment at the coordinates (2.0 nm, 2.0
nm, 7.75 nm) for each system in order to simulate the behavior of these
molecules in the gas-water interface region of nanobubbles. To begin, the
three different systems were optimized to use the least amount of energy
possible. After that, NVT molecular-dynamics simulations were carried
out for a total of 150 nanoseconds.
*2.1.2*    ***Umbrella sampling simulations***
In **Fig. S1 (b)**, the molecule of Ph, 4-HBA, or VL was placed inside the
box (their COM is (2.00 nm, 2.00 nm, 6.00 nm)), which is located directly
2.00 nm away from the COM of the water slab. The distance between the
COM of Ph, 4-HBA, or VL and that of the water slab was used as the
definition for the reaction coordinate. The weighted histogram analysis
approach, also known as WHAM, can be used to calculate the free energy
profiles of Ph, 4-HBA, or VL when they transition from the gas phase into
bulk water (Kumar et al., 1992; Hub et al., 2010). **Text S1** has an
explanation of the specifics.
*2.1.3*    ***Radial distribution function***
Estimating the strength of hydrogen bonds (HB) between specific





atoms can be done with the help of a tool known as the radial distribution
function (RDF). **Text S2** has an explanation of the specifics.
*2.2 DFT calculations*
In this work, all calculations pertaining to the electrical structure were
accomplished by utilizing the Gaussian16 program (Frisch et al., 2016). By
benchmarking at the CCSD(T)/cc-pVDZ, CBS-QB3, B3LYP/6-
311+G(d,p), MP2/6-311+G(d,p) and M06-2X/6-311+G(d,p) levels, Cao et
al. (Cao et al., 2021) found that M06-2X/6-311++G(3df,2p)//M06-2X/6-
31+G(d,p) is reliable for PhCs. Therefore, all calculations for gas-phase
reactions are performed at this level. **Text S3** contains a description of the
additional calculated details. Multiwfn (Lu and Chen, 2012) was used to
construct the electron density map. This program integrates Visual
Molecular Dynamics (version 1.9.3) (Humphrey et al., 1996) in order to
conduct an analysis of the electrostatic potential (ESP) and the average
local ionization energy (ALIE).
*2.3 IRI analysis*
Interaction Region Indicator (IRI) (Lu and Chen, 2021) was used to
determine the chemical bonds and weak interactions of Ph/4-HBA/VL
adsorbed to $TiO_2$ clusters (the details are in **Text S4**).
*2.4 Kinetic calculations*
**Text S5** contains an explanation of the kinetic calculations.
**3. Result and discussion**



### 3.1 Enrichment of Ph, 4-HBA, and VL at the A-W interface

### 3.1.1 The uptake of gaseous PhCs at the A-W interface

**Fig. S1** and **Fig. S2** illustrate the relative distributions of water, $O_3$, and PhCs molecules (Ph, 4-HBA, and VL) in the A-W interface system along the z-axis. HO$^\bullet$ are primarily situated at the A-W interface contact, with the potential to diffuse through the water slab interior (Roeselová et al., 2004). **Fig. 1(a)** displays the variation in water density along the Z-coordinate distance from 0 to 9 nm, categorizing three zones: A-W interface (2.25 to 2.79 nm and 6.21 to 6.75 nm), air (0 to 2.25 nm and 6.75 to 9 nm), and bulk water (2.79 to 6.21 nm). This method accurately determines the interfacial range (Zhang et al., 2019; Shi et al., 2020). According to location definitions, $O_3$ percentage distribution was as follows: 26% at the A-W interface; 72% in the air; and 2% in pure water (**Fig. 1(b)**). **Fig. 1(c)** depicts MD trajectories of Ph diffusion through the water slab from the air region over a 150 ns period. Ph is distributed in the air (8%) and bulk water (20%), with the majority at the A-W interface (72%) (**Fig. 1 (d)**). The majority of 4-HBA and VL molecules are located at the A-W interface, constituting 68% and 73% of the total locations as presented in **Fig. S2**.

In **Fig. 2(a)**, we observe the three key processes involving PhCs (Ph, 4-HBA, or VL) diffusing into the water slab from the air region. (I) The mutual attraction of gaseous Ph, 4-HBA, or VL and nanoparticles; (II) The uptake of PhCs (Ph, 4-HBA, or VL) at the air-nanoparticle interface; (III)





The hydration reaction of PhCs (Ph, 4-HBA, or VL) in the bulk water. **Fig.**
**2(b)** displays the free energy profile of the trajectories as Ph/4-HBA/VL
transitions from the air into the bulk water (see **Text 6** for calculations
details). The $\Delta G_{gas \rightarrow interface}$ values are $-0.22$ kcal mol $^{-1}$ for the Ph-A-W
(Phenol-Air-Water) system, $-0.45$ kcal mol $^{-1}$ for the 4-HBA-A-W (4-
hydroxybenzaldehyde-Air-Water) system, and $-0.20$ mol $^{-1}$ for the VL-A-
W (Vanillin-Air-Water) system. These values suggest that it is
thermodynamically favorable for PhCs to approach the interfacial water
molecules. **Fig. S3** illustrates typical snapshots from the trajectories of
PhCs (Ph, 4-HBA, or VL). Initially, one molecule of Ph, 4-HBA, or VL
was placed in the center of the water box, with an equivalent COM distance
of 2 nm between the PhCs and the air phase. Subsequently, the PhCs moved
closer to the interface, leading to adsorption at the A-W interface. During
the adsorption process, the H atom of phenolic hydroxyl group served as
an electron donor, binding to the oxygen atom on the surface and
preventing its return to the bulk water. Concurrently, hydrogen bonds were
formed. This property allowed the phenolic hydroxyl groups on PhCs can
effectively adhere to the A-W interface, consistent with the experimental
observations using steady-state interfacial vibrational spectra (Kusaka et
al., 2021). Based on these findings, the location where air and water meet
exhibits an increased concentration of PhCs.
***3.1.2   Interface properties of PhCs***



The research focused on understanding the behavior of PhCs at A-W
interface. The distribution probability of angle (α, β, γ) for Ph/4-HBA/VL
in relation to the A-W interface is shown in **Fig. 3(a)–(d)**. The Z-axis is
defined as the axis perpendicular to the interface. The angles are formed
between the Z-axis and the benzene ring plane, the phenolic hydroxyl
group, and the O–αC bound of Ph, 4-HBA, and VL, respectively, denoted
as α, β, and γ. In the Ph-A-W, 4-HBA-A-W, and VL-A-W systems, a broad
distribution range is observed, suggesting that PhCs are rather randomly
distributed across the interface. However, statistically, the highest
distribution range for α and β falls within 15°–25° (or 145°–165°) and
75°–95°, respectively. This applies to both α and β. In the VL-A-W system,
the highest distribution range for α is around 93°. In general, introducing
more hydrophilic functional groups increases the characteristic angle α and
β of PhCs at the interface, allowing for more secure adsorption at the water-
air interface.
To set up the interface reaction environment for further quantum
chemical calculations, the radial distribution function (g(r)) of Ph, 4-HBA,
and VL at interfaces was computed and is show in **Fig. 3 (e)–(g)**. These
figures also display the radial distribution function (RDF) and the
coordination number N of $H_{Ph-OH}$–$O_{H_2O}$, $H_{4-HBA-OH}$–$O_{H_2O}$, and $H_{VL-OH}$–$O_{H_2O}$
at the A-Winterface. Peak intensities are observed in the range of 0.25–0.3
Å for $H_{Ph-OH}$–$O_{H_2O}$, $H_{4-HBA-OH}$–$O_{H_2O}$, and $H_{VL-OH}$–$O_{H_2O}$, as shown in **Fig.**



**3(e)–(g)**, respectively. The interaction between $H_{PhCs}$ and $O_{H_2O}$ is the
primary factor influencing the stability of PhCs at the interface. The N
values of $H_{Ph-OH}–O_{H_2O}$, $H_{4-HBA-OH}–O_{H_2O}$, and $H_{VL-OH}–O_{H_2O}$ are 2.68, 2.51,
and 2.09 respectively. The number of functional groups attached to the
benzene ring affects the N value; more functional groups lead to a lower N
value. When a molecule has more functional groups, it occupied more
space and exerts a stronger repulsive force on the nearby water molecules
compared to those with fewer functional groups.
*3.2 Adsorption of Ph, 4-HBA, and VL by TiO₂ Clusters*
The investigation into the structural stability of $TiO_2$ clusters (Zhai and
Wang, 2007; Syzgantseva et al., 2011; Arab et al., 2016) has revealed six
distinct types of (($TiO_2)_n$ (n = 1–6)) clusters as depicted in **Fig. S4**. The
structural parameter values computed for $TiO_2$ clusters using the M06–
2X/6-31+G(d,p)/LANL2DZ level align with reported experimental values
(Calatayud et al., 2008; Bai et al., 2020). **Fig. 4, S5, and S6** provide
additional insights into the adsorption of PhCs on $(TiO_2)_n$ (n = 1–6) clusters.
The placement of PhCs on $TiO_2$ clusters significantly impacts adsorption
energies (Bai et al., 2020). The adsorption capacity of pollutants on cluster
surfaces is a key factor influencing degradation efficiency (Qu and Kroes,
2006). The primary mechanism of phosphorus atoms adsorption to $(TiO_2)_n$
(n = 1–4) clusters occurs at a range of 2.57 to 2.61 Å and involves
interaction between the $H_{-OH}$ atom and the $O_{TiO_2}$ atom, as seen in **Fig. 4(a)**.





Hydrogen bonds can be formed between the $H_{-OH}$ atom and the $O_{TiO_2}$ atom
(1.80–2.61 Å), improving the adsorption capacity. In contrast, Ph
adsorption to $(TiO_2)_n$ (n = 5–6) clusters, ranging from 2.08 to 2.09 Å, is
primarily due to interaction between Ti atom and $O_{-OH}$ atom. In **Fig. S5**,
the primary interaction for 4-HBA and VL occurs between the Ti atom and
the $O_{-CHO}$ atom, with distances ranging from 1.93 and 2.07 Å. TiO$_2$ clusters
have a greater potential to interact with the oxygen atom of the aldehyde
group than with the oxygen atom of the phenolic hydroxyl group. **Fig. S7**
presents the ALIE surface values for the three PhCs considered. Lower
ALIE values indicate weaker binding of electrons, with darker blue regions
signifying the lowest local minimum ALIE levels. ALIE values, in the
range of 11.42–11.97 eV, for $O_{-CHO}$ atom are lower than those for $O_{-OH}$ atom
(12.49−15.46 eV) or $O_{-OCH_3}$ atom (14.96 eV) due to the electron-accepting
nature of the aldehyde group. Therefore, the interaction between the
titanium atom and $O_{-CHO}$ atom is stronger than with the $O_{-OH}$ or $O_{-OCH_3}$
atoms.
Adsorption energy a metric of adsorption capacity, is illustrated in **Fig.**
**4(b)–(d)** for phosphorus, 4-hydroxybenzoic acid, and vinylidene
dichloride on $(TiO_2)_n$ (n = 1–6). TiO$_2$ exhibits the highest adsorption
capacity for PhCs. ($\Delta G_{ad} = -72.35$ kcal mol$^{-1}$) (**Fig. 4(b)**). The adsorption
energy difference values of TiO$_2$ and $(TiO_2)_3$ for 4-HBA and VL are −45.32
(**Fig. 4(c)**) and −102.46 kcal mol$^{-1}$ (**Fig. 4(d)**), respectively. The energy of



physorption range from −1.20 to 9.56 kcal mol $^{-1}$ illustrates the
spontaneous chemical adsorption (Nollet et al., 2003). However, the
capacity of $TiO_2$ to adsorb VL is significantly higher than that to adsorb Ph
and 4-HBA. **Fig. 4(b)–(d)** show that the adsorption capacity falls as the
size of $TiO_2$ clusters increases when n ≤ 4. In contrast, the adsorption
capacity remains constant when n > 4. IRI measurements of Ph on the
$(TiO_2)_n$ surface (**Fig. 4(e)**) reveal Ph-$TiO_2$ hydrogen bonds ($H_{Ph}$−$O_{TiO_2}$
bonds bonds) and their electrostatic and dispersion effects. Benzene C atom
exhibits $sp^2$ hybridization, meaning it forms one σ-bond and one π-bond.
The $sp^2$ hybridization of benzene explains its limited interaction with $TiO_2$
clusters and accounts for the substantial adsorption energy. Similar
interactions occur with 4-HBA and VL (**Fig. S6**). Hydrogen bonds form
between the $H_{-CHO}$ atom of 4-HBA or VL and the $O_{TiO_2}$ atom, despite the
presence of the $H_{Ph}$ atom.

### *3.3 Continuous oxidation mechanisms*

### *3.3.1   $O_3$- and HO•-initiated reactions*

PhCs, once released into the atmosphere, undergo several processes,
including adsorption on mineral aerosol surfaces, accumulation at the A-W
interface, dispersion in bulk water within liquid droplets, and oxidation
reactions initiated by atmospheric oxidants. This section delves into the
detailed mechanisms and characteristics of these reactions. At the M06-
2X/6-311++G(3df,2p)//M06-2X/6-31+G(d,p) level, the structures with the



minimum free energy for the Ph/4-HBA/VL has been determined (**Fig. S9**).
In the case of VL, a significant reduction in molecular energy is observed
due to the formation of a powerful intramolecular hydrogen bond with a
length of 2.09 Å between the H and O atoms near the methyl group.
Moreover, the lone pair electrons of oxygen atoms can form additionally
p-$\pi$ conjugations with the $\pi$ electrons of the phenyl ring, further reducing
the overall energy of VL. The statistical charts of calculated $\Delta_r G$ and $\Delta G^{\ddagger}$
values for $O_3$- and $HO^{\bullet}$-initiated reactions are displayed in **Fig. 5** and **S8**
and detailed data are available in **Tables 1–4**.
$O_3$ is a major oxidant in the atmosphere, with high concentrations in
the troposphere ranging between $7 \times 10^{11}$ molecules cm$^{-3}$ (Platt et al., 1984;
Prinn et al., 1995b). Investigating the fate of PhCs in the presence of $O_3$ is
essential. The ozonolysis of PhCs involves the synthesis of primary
ozonide, the formation of active Criegee intermediate (CI), and the
disintegration of CI. The $O_3$-initiated reactions of Ph/4-HBA/VL involve
radical adduct formation (RAF) channels on the benzene ring ($R_{O_3\text{-RAF}}1-$
6), highlighted in red in **Fig. S9**. **Fig. 5(a)–(d)** depict that the ozonolysis
pathways $R_{O_3\text{-RAF}}$ are exergonic, indicating their spontaneity. The average
$\Delta G^{\ddagger}$ values for the ozonolysis of Ph/4-HBA/VL are ranked as Ph > VL >
4-HBA. The following is a list of the average values for the ozonolysis of
Ph/4-HBA/VL, as illustrated in **Fig. 5(e)**–**5(h)**, Ph is superior to VL and 4-
HBA, with the exception on $TiO_2$ clusters. **Fig. 5(e)** illustrates that the



average value of $\Delta G^{\ddagger}$ for $O_3$ + Ph reactions at the A-W interface is 15.38
kcal mol$^{-1}$, the lowest value out of the three PhCs. The average $\Delta G^{\ddagger}$ values
for the ozonolysis of Ph/4-HBA/VL are as follows: VL (13.95 kcal mol$^{-1}$)
< Ph (24.70 kcal mol$^{-1}$) < 4-HBA (25.16 kcal mol$^{-1}$) on TiO$_2$ clusters (**Fig.**
**5(f)**). The average $\Delta G^{\ddagger}$ values for $O_3$ + VL reactions in gas phase are the
highest among the four different EM (23.28 kcal mol$^{-1}$) shown in **Fig.**
**5(g)**). Comparing the phenolic oxidation in each of these four EM (bulk
water, interface, TiO$_2$ clusters, and gas phase) reveals that A-W interface
are more conducive to the ozonolysis of Ph, whereas TiO$_2$ clusters are more
conducive to the ozonolysis of VL. The effect of solvation on $\Delta G^{\ddagger}$ is
predominantly caused by the hydration of the phenolic OH group, as this
is the part of the molecule being dissolved. However, the presence of water
molecules in the region around the phenyl group has been shown to have a
considerable influence on the $\Delta G^{\ddagger}$ values.
HO$^{\bullet}$, known as "atmospheric detergents", is another significant
atmospheric oxidant (Atkinson, 1986; Zhang et al., 2020). According to
research by Prinn (Prinn et al., 1995a), the worldwide average
concentration of HO$^{\bullet}$ during 12 hours per day is roughly $1.6 \times 10^6$
molecules cm$^{-3}$. For this reason, elucidating the reaction mechanism
underlying $O_3$ + PhCs reactions in the troposphere is of the utmost
importance. HO$^{\bullet}$-initiated reaction pathways of Ph/4-HBA/VL include
RAF, hydrogen atom abstraction (HAA) channels from the benzene ring



($R_{HAA}$ben1–6) and the substituent group ($R_{HAA}$sub7–9). Previous research
(Gao et al., 2019) has shown that the process of single electron transfer
(SET) does not significantly contribute to the HO˙-initiated reactions
examined. Once the hydroxyl adducts or $H_2O$ are formed, significant heats
(4.21–30.28 kcal mol $^{-1}$) are released (**Fig. 5(i)–(l), S8 (a)–(d) and (i)–(l);**
the detail data in **Table S3**), indicating high thermodynamic feasibility. The
average $\Delta G^{\ddagger}$ values for HO˙-initiated reactions (**Fig. 5(m)–(p), S8 (e)−(h)**
and **(m)–(p)**) are lower than those for $O_3$-initiated reactions. Routs
$R_{HAA}$ben make a minimal contribution to HO˙-initiated reactions. At the A-
W interface, VL (3.52 kcal mol $^{-1}$) < Ph (4.52 kcal mol $^{-1}$) < 4-HBA (9.50
kcal mol $^{-1}$), and the $\Delta G^{\ddagger}$ value of Ph is the lowest (-0.97 kcal mol $^{-1}$), the
case for pathways $R_{RAF-HO}$• (**Fig.5(m)**). Among the three aromatic
compounds, the $R_{RAF-HO}$•  routes of VL on $TiO_2$ clusters are the most
favorable (**Fig. 5(n)**). When compared to HO˙-initiated reactions of
aromatic compounds in the gas phase (**Fig. S8(e)**) or bulk water (**Fig. S8(f)**),
the process of Ph + HO˙ reactions at the A-W interface is accelerated,
whereas the process of VL + HO˙ reactions is accelerated by $TiO_2$ clusters.
These findings are in agreement with the ozonolysis findings. The same
guidelines can be used to routes $R_{HAA}$sub (**Fig. 5(o), (p), S8 (g) and (h)**)
and $R_{HAA}$ben (**Fig. S(m)−(p)**). The following is a ranking of the average
$\Delta G^{\ddagger}$ values for routes $R_{RAF-HO}$• in the gas phase or bulk water: Ph < 4-HBA
< VL. As a result of having the lowest $\Delta G^{\ddagger}$ values among all HO˙-initiated





reaction mechanisms, routes $R_{RAF}$ are the most advantageous of all the
possible reaction mechanisms. In light of this, each and every route $R_{RAF-}$
$_{HO}$• and $R_{HAA}$sub will be dissected in detail.
**Fig. 6** shows the $\Delta_r G$ and $\Delta G^{\ddagger}$ values of $O_3$- and HO•-initiated reactions
at various reaction locations. These reactions are almost entirely
exothermic, with a close correlation between $\Delta_r G$ values and $\Delta G^{\ddagger}$ values.
The $\Delta G^{\ddagger}$ values for the Phe + $O_3$ reactions shown in **Fig. 6(a)** are the lowest
among the three compounds, ranging from −0.97 to 7.86 kcal mol $^{-1}$.
Exergonic and spontaneous addition reactions took place at the C1−C2 and
C3–C4 locations of Ph and VL, respectively. Because of their low $\Delta G^{\ddagger}$
values, the C1–C2 and C2–C3 sites of $O_3$-initiated reactions for 4-HBA are
advantageous. Their values are 21.76 and 22.03 kcal mol $^{-1}$, respectively.
The C1–C2 location of 4-HBA is activated to a greater extent at the A-W
interface in comparison to the gas phase and bulk water. However, the $\Delta G^{\ddagger}$
values of $O_3$ + Ph reactions on $TiO_2$ clusters are significantly greater than
those of the A-W interface (12.86−18.10 kcal mol $^{-1}$) than 24.30−25.34
kcal mol $^{-1}$. The VL + $O_3$ reactions on $TiO_2$ clusters are favorable at the
C2−C3 and C4−C5 locations (the $\Delta G^{\ddagger}$ values are 11.42 and 11.14 kcal mol
$^{-1}$, respectively, **Fig. 6(b)**). This can be explained by the fact that the
electron cloud has a greater propensity to congregate in the places C2−C3
and C4−C5, respectively. In addition, the p orbitals of the methoxy and
hydroxy groups are conjugated to the benzene ring, which offers a



powerful electron-donating conjugation effect (Aracri et al., 2013).
Because of this, the oxidation of aromatic molecules is thermodynamically
more favorable than the oxidation of the aldehyde group. Clearly, the $\Delta G^{\ddagger}$
values of HO$^{\bullet}$-initiated reactions (-0.97–13.46 kcal mol $^{-1}$) in **Fig. 6(c)–(f)**
are lower than those of O$_3$-initiated processes (11.14–27.83 kcal mol $^{-1}$) at
different points in A-W interface and TiO$_2$ clusters. This can be seen by
comparing the values to each other. At the A-W interface, the most
advantageous position for the phenol hydroxyl group to be in for Ph/4-
HBA/VL + HO$^{\bullet}$ reactions are the ortho position (**Fig. 6(c)**). OESI-MS,
which stands for online electrospray ionization mass spectrometry, was
also able to identify the hydroxylation product known as 3,4-
dihydroxybenzaldehyde (Rana and Guzman, 2020). In **Fig. 6(d)**, the ortho-
and meta-sites of phenol hydroxyl are, respectively, the most favorable
positions for Ph/4-HBA + HO$^{\bullet}$ reactions on the TiO$_2$ clusters. On the other
hand, all of the VL sites on the TiO$_2$ clusters are advantageous. At the A-W
interface and on the TiO$_2$ clusters, the abstraction of hydrogen atoms
follows the order of H$_{CHO}$ atom > H$_{OCH_3}$ atom > H$_{OH}$ atom in **Fig. 6(e)**
**and (f)**. This can also be explained by the ALIE values of these atoms listed
in the same order of H$_{CHO}$ atom (11.67–11.74 eV) > H$_{OCH_3}$ atom (14.06
eV) > H$_{OH}$ atom (15.46 eV), as shown in **Fig. S7**.
### 3.3.2 Generation and degradation of key products





For the purpose of this discussion, the primary atmospheric destiny of
the selected aromatic compounds was taken into consideration to be their
bimolecular reactions with $O_2/O_3$. **Fig. 7** and **S10** illustrate the subsequent
reaction mechanisms of IMs, respectively. $IM_{1-2}$ was produced using the
pathway that offered the best conditions for the $HO^{\bullet}$-initiated reaction of
Ph. As can be seen in **Fig. 7(a)**, the addition of $O_2$ to the C3 sites of the
$C_6H_5O$ radicals results in the formation of $C_6H_5O$-OO with no barriers in
either the gas phase or the bulk water. This is a desirable outcome. For the
transformation of the $C_6H_5O_2$-OO radicals that were created, the ring
closure reaction to form $C_6H_5O_2$-OO-d is the most attractive option.
However, it must overcome an energy barrier of 18.83 kcal mol $^{-1}$ in the
gas phase or 13.67 kcal mol $^{-1}$ in bulk water. The $C_6H_5O_2$-OO-$d_1$ radical,
which was produced by the $C_6H_5O_2$-OO-d reaction, interacts once more
with $O_2$. Malealdehyde (P1) is what should mostly result from the reaction
of the $C_6H_5O_2$-OO-$d_1$ radical with NO. However, during this process, it still
needs to overcome an energy barrier of 49.5 (in the gas phase) or 50.83
kcal mol $^{-1}$ (in the bulk water) to generate $C_6H_5O_2$-OO-$d_3$ radical; as a
result, the further transformation of the formed $C_6H_5O_2$-OO-$d_2$ should
continue very slowly. Pyrocatechol (P2) is the primary product generated
in the gas phase and bulk water when the H atom of the $C_6H_5O_2$-OO radical
is displaced. At the A-W interface, a sequence of hydroxylation products,
including pyrocatechol (P2), benzene-1,2,3-triol (P3), and benzene-





1,2,3,4,5,6-hexaol (P4), are generated through hydroxylation processes
rather than by a single SET ($\Delta G^{\ddagger}$ = 111.79 kcal mol$^{-1}$). OESI-MS was also
able to identify these hydroxylation products (Rana and Guzman, 2020). In
order to gain a more comprehensive understanding of the reaction
mechanism at the A-W interface, the major product (the $C_7H_5O_2$ radical)
for pathways $R_{HAA}$ of 4-HBA was also taken into consideration. According
to **Fig. S10(a)**, the addition of HO$^{\bullet}$ to the C7 sites of the $C_7H_5O_2$ radical
can occur without any obstructions. The overpowering of the 18 kcal mol
$^{-1}$ barrier resulted in the formation of the hydroxylation products (4-
hydroxybenzoic acid (P5), 3,4-dihydroxybenzoic acid (P6), 2,3,4-
trihydroxybenzoic acid (P7), and 2,3,4,5,6-pentahydroxybenzoic acid
(P8)). There was found to be one transition route for the continued
ozonolysis of the hydroxylation products that were produced in P6. The
C2–C3 site of P6 to create P6-5$O_3$ ($\Delta G^{\ddagger}$ = 16.59 kcal mol$^{-1}$) has the lowest
activation energy of all the available paths for the relevant reactions (**Fig.**
**S10(b)**). This corresponds to a value of 16.59 kcal mol$^{-1}$. When the $\Delta G^{\ddagger}$
values of the breakage of five-membered rings created by ozonolysis
pathways are compared, one can get the conclusion that the formation of
$IM_{P6}$-5$O_3$-a is the most favored pathway. All of the hydrogen abstraction
processes involving $H_2O$ and $IM_{P6}$-5$O_3$-a have rather high energy barriers
(32.93 kcal mol$^{-1}$). On the other hand, the very low $\Delta G^{\ddagger}$ values of the -
NO-O abstraction make it a desirable choice. Following a chain of





ozonolysis reactions, the following products were obtained: ((2E,4Z)-2-
formyl-4,5-dihydroxy-6-oxohexa-2,4-dienoic    acid    (P9);    2,3-
dihydroxymalealdehyde (P10); and 2,3-dioxpropanoic acid (P11).
Therefore, the product that was created, P10, may also be the product that
was discovered through experimentation (mass to charge ratios (m/z) = 115)
(Rana and Guzman, 2020).
**3.4 Comparison with available experimental results**
The rate constants ($k$) of the overall reaction under the temperature
range of 278–318 K were computed based on acquired potential energy
surfaces for the $O_3$-initiated and $HO^\bullet$-initiated reactions of selected
compounds. The results of these calculations are listed in **Table S5 and S6**,
respectively. The temperature dependences of the various $k$ values for Ph,
4-HBA, and VL at the A-W interface and in bulk water are depicted in **Fig.**
**8**. At low values of $k$, there is a positive dependence on temperature. When
the $k$ values are raised to a certain degree, the temperature dependency
seems to lose any significance it may have had before. The following is an
order of the $k$ values for $O_3$-initiated reactions: $Ph_{A-W} > VL_{TiO_2} > VL_{A-W} >$
$4\text{-}HBA_{A-W} > 4\text{-}HBA_{TiO_2} > Ph_{TiO_2}$ (**Fig. 8(a)**). According to **Fig. 8(b)**, the $k$
values of $HO^\bullet$-initiated reactions go as follows: $VL_{TiO_2} > Ph_{A-W} > VL_{A-W} >$
$4\text{-}HBA_{TiO_2} > Ph_{TiO_2} > 4\text{-}HBA_{A-W}$. If we look at **Fig. 8(a)** and **Fig. 8(b)**, we
can see that the $k$ values of $HO^\bullet$-initiated reactions are one hundred times
greater than those of $O_3$-initiated reactions. **Table 1** is a listing of the





experimental and estimated k values that are available for $O_3$-initiated and
$HO^\bullet$-initiated reactions at 298 K. According to the findings, the ozonolysis
of Ph was promoted by the water-gas interface as well as by $TiO_2$ clusters,
and the $HO^\bullet$ initiated reactions of VL were promoted by $TiO_2$ clusters.
However, the $O_3/HO^\bullet$ + 4-HBA reactions have the lowest $k$ values among
the three molecules when tested in a variety of environmental
environments. The estimated $k_{O_3+Ph}$ values at the A-W interface are 11
orders of magnitude greater than those of catechol under dry conditions in
gas phase (Zein et al., 2015), when compared with the experimental data.
Because it has a higher $k_{O_3}$ value, catechol, which is one of the main
products of Ph's oxidation in the atmosphere, has a higher degree of
reactivity than its parent compound (**Table 1**). The estimated value of VL
is lower than the experimentally determined value of $k_{O_3}$ for guaiacol under
dry conditions, which is $(0.40 \pm 0.31) \times 10^{-18}$ $cm^3$ molecule $^{-1}$ s $^{-1}$ in the
gas phase (Zein et al., 2015). The difference between the predicted value
of $k_{HO^\bullet+VL}$ is $1.14 \times 10^{-10}$ $cm^3$ molecule $^{-1}$ s $^{-1}$ and the average experimental
value of $k_{HO^\bullet}$ for methoxyphenols is just an order of magnitude. As a
consequence, the findings of our calculations are reliable.
***3.5 Ecotoxicity assessment***
We made predictions about the ecotoxicity of Ph, 4-HBA, and VL to
three different trophic levels of aquatic creatures (fish, daphnia, and green
algae) in order to better understand how the atmospheric oxidation process



affects aquatic organisms, specifically fish, daphnia, and green algae. The
acute toxicity of Ph, 4-HBA, and VL to aquatic organisms follows the order
indicated in **Fig. S11(a)**, which is "green algae > daphnid > fish."
According to the criteria in **Table S7**, the acute toxicity of Ph, 4-HBA, and
VL is either "very toxic" or "toxic" for three aquatic organisms at
concentrations ranging from 2.40–27.70 mg L$^{-1}$. The transformation
products have a greater average acute toxicity dosage to three aquatic
creatures than their parent chemicals did (0.79–1.33 mg L$^{-1}$), as shown by
the fact that the transformation products have a value of 1.40–2.82 mg L$^{-1}$.
On the other hand, the acute toxicity of some products, such as P1 (0.26–
1.00 mg L$^{-1}$) and P10 (0.54 mg L$^{-1}$ to "green algae"), is higher than that
of their parent chemicals. On the other hand, **Fig. S11(b)** demonstrates that
the sequence of "green algae > fish > daphnid" is the one that has the
highest average chronic toxicity. In addition, the longterm toxicity of
transformation products is often detrimental, but it is lower than that of the
parent chemicals. On the other hand, the chronic toxicity of P1, P2, P3, and
P11 is still "toxic/very toxic" to green algae, fish, and daphnid.
Consequently, there remains a concern regarding the potential hazards
associated with certain transformation products.
**4. Conclusions**
Combining molecular dynamic simulations (with the AMBER force
field) and quantum chemical calculations (at the M06–2X/6–



311++G(3df,2p)//M06–2X/6-31+G(d,p) level) methods has provided

comprehensive insights into the surface properties of Ph, 4-HBA, and VL,

as well as their reactions induced by $O_3$ and $HO^{\bullet}$, both in homogeneous and

heterogeneous environments. Here are some key findings from this

research:

(1) Free energy well of Ph, 4-HBA, and VL favor the A-W interface as

their preferred location, with the occurrence percentages of approximately

~72%, ~68%, and ~73% respectively. Ph and 4-HBA show a preference for

the A-W interface over the air, with energy difference of around 0.22 and

0.45 kcal mol $^{-1}$. The VL adsorbed on the $TiO_2$ clusters has a higher

likelihood of remaining compared to VL adsorbed at the A-W interface. (2)

The adsorption capacity of $TiO_2$ clusters decreases with increasing cluster

size until n > 4. After that point, the adsorption capacity remains constant.

Strong electrostatic attractive interactions and attractive dispersion effects

occur between the benzene of the Ph and Ti atoms. Hydrogen bonds form

between the atom of $O_{TiO_2}$ and the $H_{-CHO}$ group of 4-HBA or VL. (3) The

$O_3$- and $HO^{\bullet}$-initiated reactions for Ph and VL are facilitated by the A-W

interface and $TiO_2$ clusters, respectively, For $O_3$-initiated reactions at the

A-W interface, the C1–C2 position on the benzene ring is most favorable.

In both the A-W interface and on $TiO_2$ clusters, the total branching ratio for

routes $R_{RAF}$ and $R_{HAA}$sub is 72.68% ~ 100%. For route $R_{HAA}$sub, the order

is $H_{-CHO}$ atom > $H_{-OCH_3}$ atom > $H_{-OH}$ atom. (4) The $k$ values (in



molecules·cm$^{-3}$ s$^{-1}$, at 298K and 1 atm) of $O_3$-initiated reactions follow
the order of $Ph_{A-W}$ (5.98 × 10$^{-7}$) > $VL_{TiO_2}$ (3.30 × 10$^{-15}$) > $VL_{A-W}$ (1.27 × 10
$^{-17}$) > 4-HBA$_{A-W}$ (6.79 × 10$^{-23}$) > 4-HBA$_{TiO_2}$ (5.32 × 10$^{-24}$) > $Ph_{TiO_2}$ (1.84
× 10$^{-24}$). The $k$ values of HO$^•$-initiated reactions follow the order of $VL_{TiO_2}$
(6.70 × 10$^{-6}$) > $Ph_{A-W}$ (2.69 × 10$^{-6}$) > $VL_{A-W}$ (1.73 × 10$^{-7}$) > 4-HBA$_{TiO_2}$
(3.16 × 10$^{-9}$) > $Ph_{TiO_2}$ (3.17 × 10$^{-10}$) > 4-HBA$_{A-W}$ (9.49 × 10$^{-11}$). (5) Toxicity
risk assessment on aquatic species reveal that most of the reaction products
are significantly less harmful than the parent compounds. However,
products P1, P2, P3, P10, and P11 are more hazardous, and further
investigation of their atmospheric fate is recommended.
Ph undergoes transformation to malealdehyde and catechol when
exposed to $O_3$ or HO$^•$ in the troposphere. When Ph/VL is at the droplet
aerosol interface, rapid oxidation to polyhydroxylated compounds occurs.
VL eventually creates tiny molecule aldehydes and acids. This oxidation
process is accelerated when VL is encased in a mineral aerosol represented
by $TiO_2$ clusters. It is recommended that enterprises producing lignin, such
as those in the pulp and paper industry, or factories that employ lignin in
the manufacturing of adhesives, rust inhibitors, color dispersants, diluents,
or other similar products, be constructed in regions with low relative
humidity. It is recommended that treatment facilities that collect lignin
pyrolysis products and recycle the byproducts be located in the surrounding
area.



**Data availability**

Data related to this article are available online at https://doi.org/10.5281/zenodo.10614650.

**Author contributions**

Yanru Huo contributed to the manuscript conceptualization, methodology, software, formal analysis, investigation, and writing of the original manuscript. Mingxue Li provided insight into the writing ideas throughout the article. Xueyu Wang offered some guidance on the method section of the manuscript. Jianfei Sun, Yuxin Zhou, and Ma Yuhui reviewed the original manuscript. Maoxia He: Conceptualization, Resources, Writing – review & editing, Supervision, Funding acquisition.

**Competing interests**

The contact author has declared that none of the authors has any competing interests.

**Acknowledgements**

This work was financially supported by the National Natural Science Foundation of China (NSFC No. 22276109, 21777087, and 21876099).

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





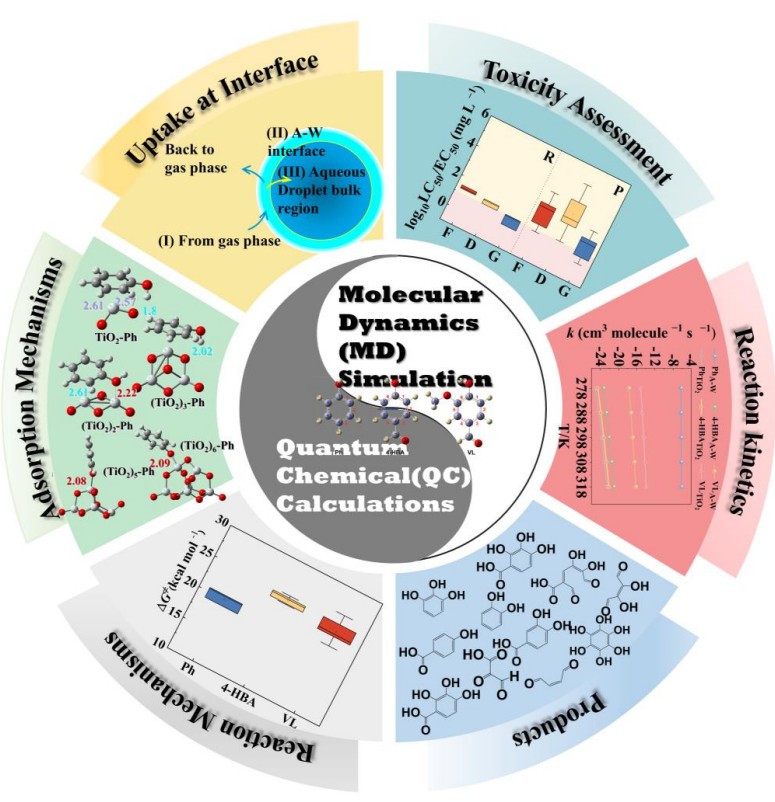

*Graphical Abstract*














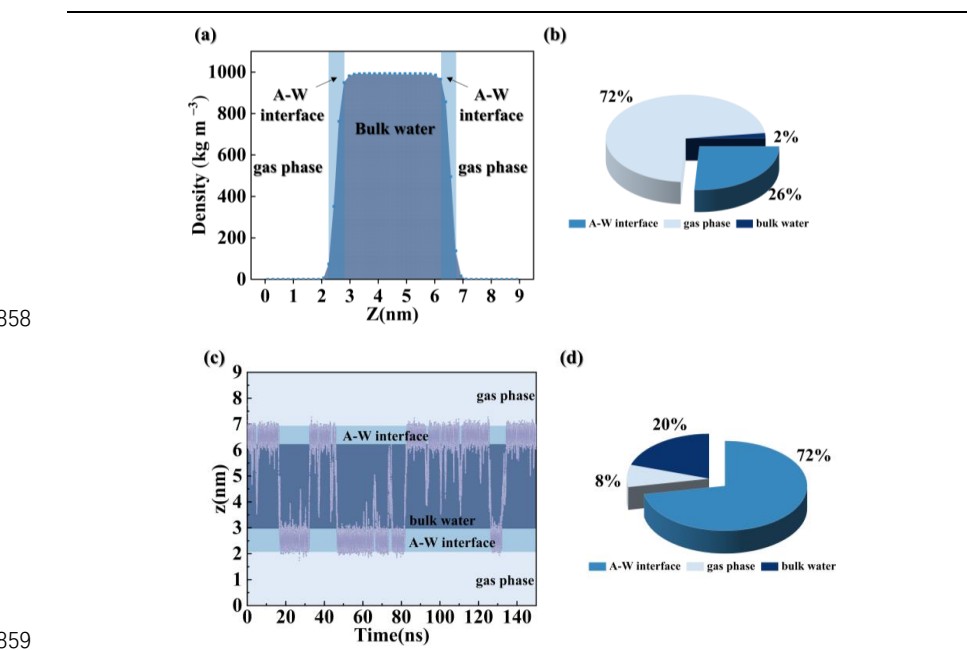



**Fig.1 (a)** Relative concentration distributions in the A-W system along the z-axis; **(b)** probability of
$O_3$ at the A-W interface, in gas phase, and in bulk water; **(c)** MD trajectories of Ph diffusion through
the water slab over a 150 ns period; **(d)** probability of Ph at the A-W interface, in gas phase, and in
bulk water.

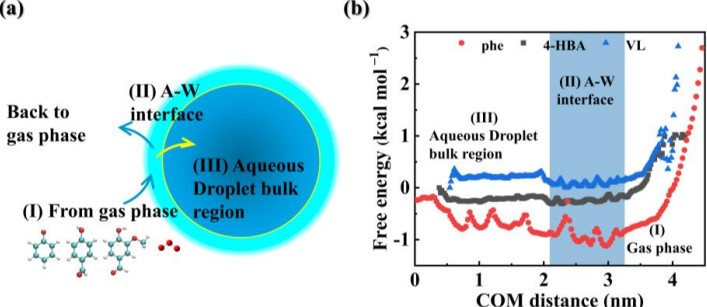


**Fig.2 (a)** Three key processes for the reaction of gaseous PhCs (Ph, 4-HBA, or VL) with the
nanoparticles; **(b)** free energy profile of gaseous PhCs (Ph, 4-HBA, or VL) approaching the bulk
water.

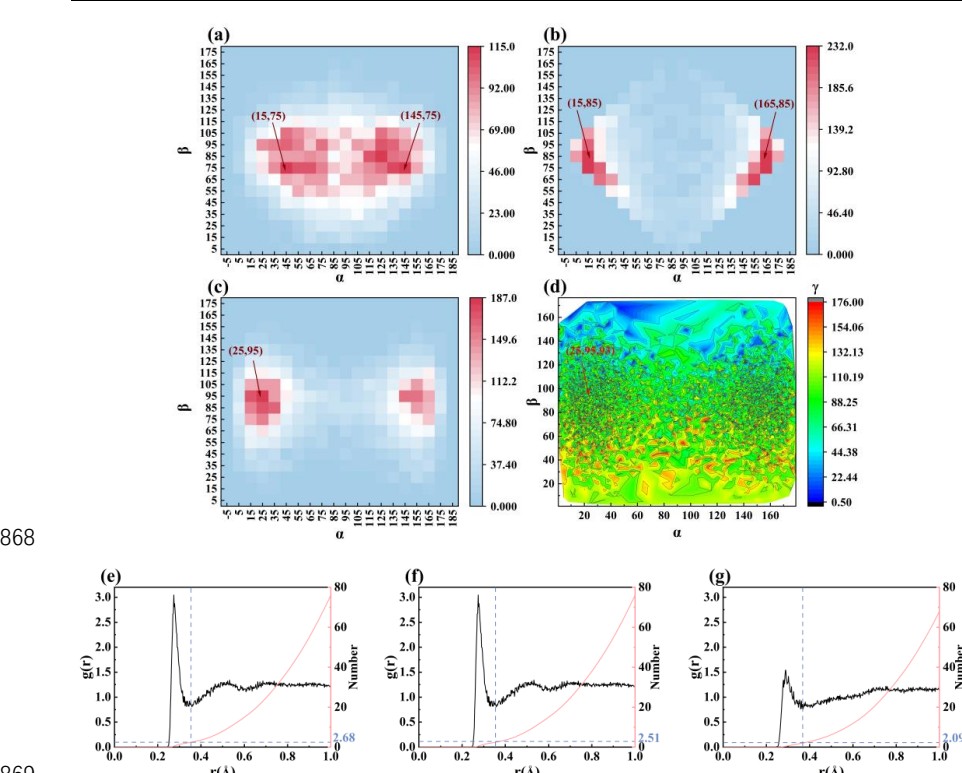



**Fig. 3** Angle (α, β, γ) distribution probability of **(a)** Ph, **(b)** 4-HBA, or **(c)** VL with respect to A-W
interface; radial distribution function (RDF) and the coordination number $N$ of **(e)** $H_{Ph-OH}$-$O_{H_2O}$, **(f)**
$H_{4-HBA-OH}$-$O_{H_2O}$, and **(g)** $H_{VL-OH}$-$O_{H_2O}$ at the A-W interface.






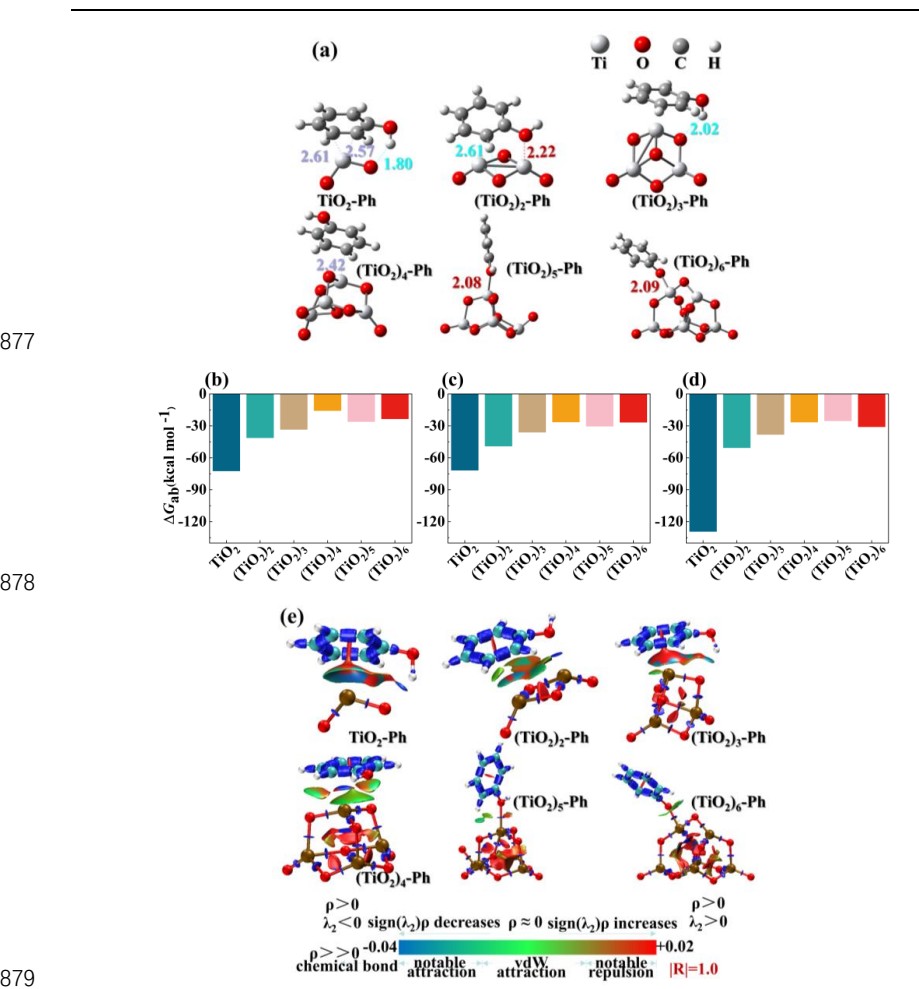

**Fig. 4** Adsorption details of PhCs on $(TiO_2)_n$ (n = 1−6) clusters; **(a)** structure of Ph adsorption on

$(TiO_2)_n$ (n = 1−6) surface; adsorption energy of **(b)** Ph, **(c)** 4-HBA, and (d) VL on $(TiO_2)_n$ (n = 1−6,

unit: kcal mol$^{-1}$); **(e)** Interaction region indicator (IRI) analyses of Ph on $(TiO_2)_n$ (n = 1−6) surface.



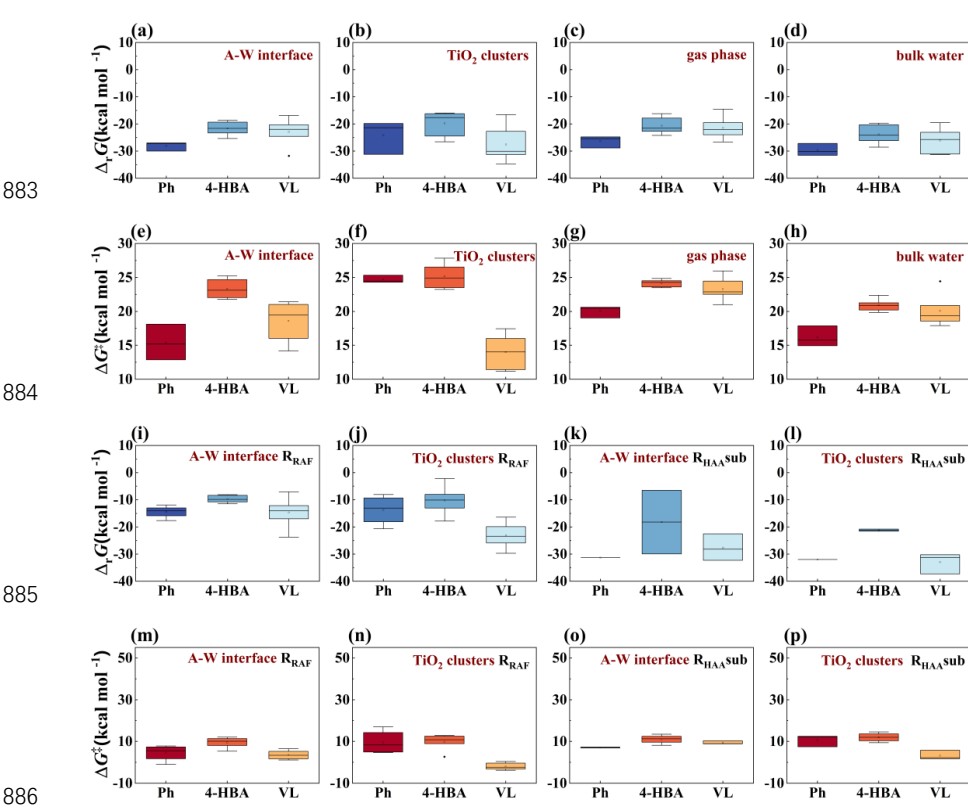

**Fig. 5** Statistical charts of calculated **(a)**–**(d)** $\Delta_r G$ and **(e)**–**(h)** $\Delta G^{\ddagger}$ values for $O_3$-initiated reactions;

**(i)**–**(l)** $\Delta_r G$ and **(m)**–**(p)** $\Delta G^{\ddagger}$ values for $HO^{\bullet}$-initiated reactions.



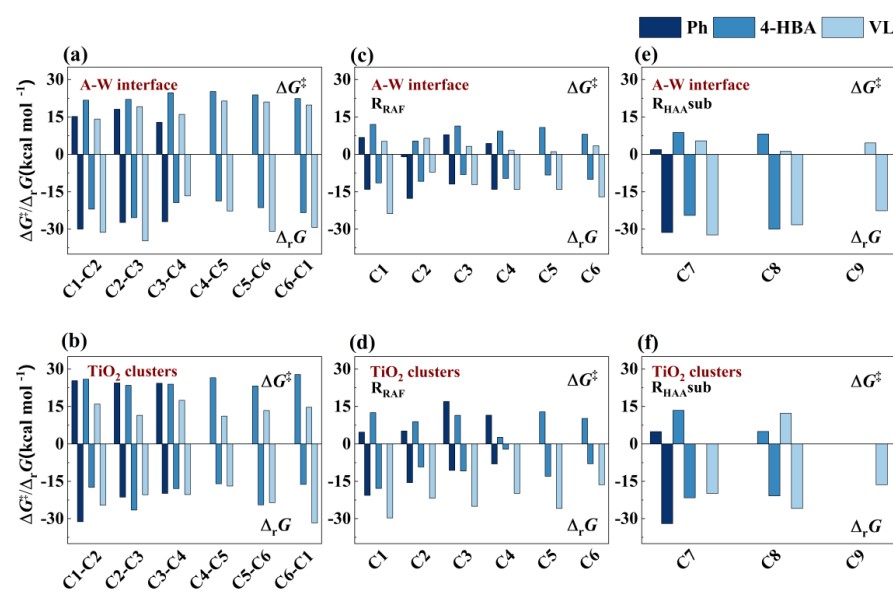


**Fig.6** $\Delta_r G$ and $\Delta G^\ddagger$ values of **(a)**–**(b)** $O_3$-initiated reactions and **(c)**–**(f)** HO•-initiated reactions at
different reaction positions.





**(a)**


**(b)**


**Fig.7** Subsequent reaction mechanisms of important intermediates (IMs) (unit: kcal mol $^{-1}$) in **(a)**

gas phase / bulk water and at **(b)** A-W interface.





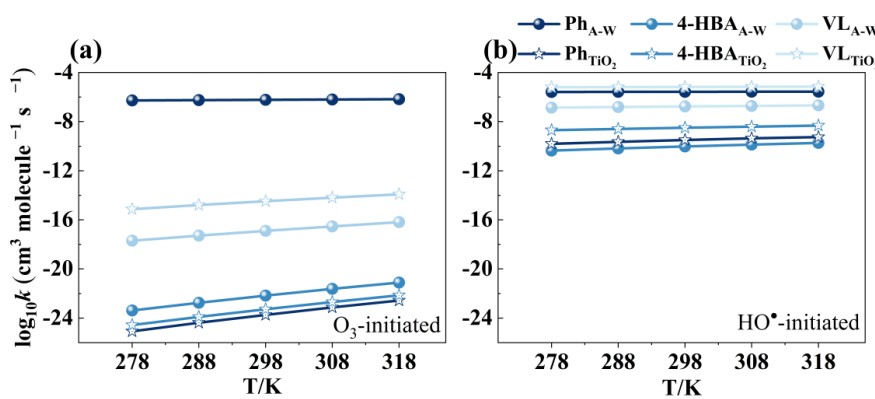

**Fig.8** Calculated rate constants for the initial reactions of Ph, 4-HBA, and VL with $O_3$ and $HO^\bullet$ at

different temperatures (278–318 K) and 1 atm.





**Table 1** The available experimental and calculated reaction rate constants ($k$) values of $O_3$-
initiated and HO•-initiated reactions at 298 K. Unit: $cm^3$ molecule$^{-1}$ s$^{-1}$.

| Compounds | $k_{tot\text{-}A\text{-}W,cal}$[a] | $k_{tot\text{-}TiO_2,cal}$[b] | $k_{tot\text{-}gas,cal}$[c] | $k_{tot\text{-}wat,cal}$[d] | $k_{,exp}$ | Ref. |
|---|---|---|---|---|---|---|
| Ph | $5.98 \times 10^{-7}$ | $1.84 \times 10^{-24}$ | $5.27 \times 10^{-20}$ | $4.02 \times 10^{12}$ | $(13.5 \pm 1.1) \times 10^{-18,\,e}$ | Zein et al. (2015) |
|  | $2.69 \times 10^{-6}$ | $3.17 \times 10^{-10}$ | $2.34 \times 10^{-9}$ | $4.46 \times 10^{13}$ | — |  |
| 4-HBA | $6.79 \times 10^{-23}$ | $5.32 \times 10^{-24}$ | $4.93 \times 10^{-24}$ | $1.97 \times 10^{12}$ | — |  |
|  | $9.49 \times 10^{-11}$ | $3.16 \times 10^{-9}$ | $7.90 \times 10^{-11}$ | $2.52 \times 10^{13}$ | — | Rana et al. (2020) |
| VL | $1.27 \times 10^{-17}$ | $3.30 \times 10^{-15}$ | $1.35 \times 10^{-22}$ | $2.20 \times 10^{12}$ | $(0.40 \pm 0.31) \times 10^{-18,\,f}$ | Zein et al. (2015) |
|  | $1.73 \times 10^{-7}$ | $6.70 \times 10^{-6}$ | $1.14 \times 10^{-10}$ | $3.15 \times 10^{13}$ | $6.00 \times 10^{-11,\,g}$ | Rana et al. (2020) |

[a]: calculated values of phenolic compounds at A-W interface;
[b]: calculated values of phenolic compounds on $TiO_2$ clusters;
[c]: calculated values of phenolic compounds in the gas phase;
[d]: calculated values of phenolic compounds in the bulk water.
[e]: experimental values of catechol in the gas phase;
[f]: experimental values of guaiacol in the gas phase;
[g]: experimental average $k_{HO•}$ values of methoxyphenols in the gas phase.