# Peer review of "Measurement report: Rapid oxidation of phenolic compounds by O₃ and HO•: effects of air-water interface and mineral dust in tropospheric chemical processes"

_EGUsphere, 2023_

## Author Comment (AC1)

**MS No.:** egusphere-2023-2856

**TITLE:** *Measurement report: Rapid oxidation of phenolic compounds by O₃ and HO•:*

*effects of air-water interface and mineral dust in tropospheric chemical processes*

Dear editor and reviewers,

Thank you for your comments on our manuscript to enrich the content of this work. We have tried our best to modify the manuscript according to your suggestions. Each question was answered in blue; every correction is in red and the deleted texts are marked with a delete line number in this response. Besides, every correction is in red in the revised manuscript. The detailed response to each comment is shown as follows:

**Reviewer #1:** The authors applied a combination of molecular dynamics simulation and quantum chemical calculations to systematically investigate the atmospheric transformation processes and environmental impacts in different environmental media of three phenolic compounds (PhCs). In addition, the authors assessed the ecological exposure risk of phenol (Ph), 4-hydroxybenzaldehyde (4-HBA), and vanillin (VL) and their transformation products based on predicted toxicological data. The results are important for well understanding the transition and fate the gas-phase reaction processes of PhCs. The expression and analysis of this paper are intelligible and logical. Nevertheless, the current manuscript still has several areas for improvement. The authors are advised to revise the manuscript according to the following suggestions.

1.  Why did you choose phenol (Ph), 4-hydroxybenzaldehyde (4-HBA), and vanillin (VL) as model compounds? It is better to explain in the introduction.

*Author reply*: Thank you for your suggestions. The reason why the model compounds were selected has been added to line 104: "Increasing the number of constituents on the aromatic ring would affect the reactivity and lead to complex compounds after reaction addition and/or open ring pathways".

2. In calculation methods, it is required for illustrating the reliability of the chosen methods and the used models. Moreover, the possible error bars should be further discussed. For example, in gas phase kinetics calculations, the recrossing effects, anharmonicity, and torsional anharmonicity are not considered. How do these factors influence the calculated results? In addition, what is the possible accuracy for electronic structure calculations?

*Author reply*: Thank you for your questions. This study was conducted under multiple atmospheric ambient media conditions, not just in the gas phase environment. In order to compare the kinetic data under different ambient media conditions, we used the same method for rate constant calculations under different ambient media conditions. Moreover, our calculated the reaction rate constants for the ozonolysis of Ph is $5.27 \times 10^{-20}$ cm$^3$ molecule$^{-1}$ s$^{-1}$ which is less than one order of magnitude different from the reaction rate constants for the ozonolysis of guaiacol. There is no need to consider torsional anharmonicity because PhCs contain benzene rings. Therefore, our results are reliable.

The benchmark of electronic structure calculations in Lines 165-171:

"In this work, all structural optimization and energy calculation were accomplished by utilizing the Gaussian16 program (Frisch et al., 2016). By benchmarking at the

CCSD(T)/cc-pVDZ, CBS-QB3, B3LYP/6-311+G(d,p), MP2/6-311+G(d,p) and M06-2X/6-311+G(d,p) levels, Cao et al. (Cao et al., 2021) found that M06-2X/6-311++G(3df,2p)//M06-2X/6-31+G(d,p) is reliable for PhCs at gas phase. Therefore, all calculations for gas-phase reactions are performed at this level".

Cao et al. (Cao et al., 2021) reported that the O−H bond dissociation energies are 89.2, 85.8, 81.9, 114.5, and 87.4 kcal mol$^{-1}$ for CCSD(T)/cc-pVDZ, CBS-QB3, B3LYP/6–311 + G(d,p), MP2/6–311 + G(d,p) and M06-2X/6–311 + G(d,p), respectively. The result implies that M06-2X/6–311 + G(d,p) level produces reliable results for phenol.

3. Have you scaled the calculated frequencies? Have you considered the stability of wave function in DFT calculations?

*Author reply*: Yes. The frequency correction factor (0.967) has been taken into account. After analysing the stability of the wavefunction, the method we used is reliable.

```
 40  / 41            0.5565
 40 -> 44            0.20599
The wavefunction is stable under the perturbations considered.
 Leave Link  914 at Sat Jun 15 11:12:18 2024, MaxMem= 17179869184 cpu:
 (Enter /home/hmx/program/g16/l601.exe)
 Copying SCF densities to generalized density rwf, IOpCl= 0 IROHF=0.
```

4. It is necessary for explaining that TiO$_2$ clusters can be used to model the mineral dust in the atmosphere. It is well known that there is abundant water in the atmosphere. Do TiO$_2$ clusters react with water?

*Author reply*: The significance of using TiO$_2$ clusters to model the mineral dust in the atmosphere has already been established in the introduction. Line 96-103, "In chemical engineering, titanium dioxide (TiO$_2$) is an essential photoactive component found in atmospheric mineral dust (Sakata et al., 2021; Wang et al., 2023). The interaction

between PhCs and TiO$_2$ is continuous (Grassian, 2009; Rubasinghege et al., 2010; Shang et al., 2021), despite the relatively low prevalence of TiO$_2$ mineral particles (comprising 0.1% to 10% by mass). Therefore, it is essential to investigate the disparity in the oxidation reaction mechanisms and kinetics of PhCs at A-W interface and mineral dust particles."

Only the reaction of PhCs with HO$^\bullet$ was considered in this manuscript. TiO$_2$ is a semiconductor material (the forbidden band width (E$_g$) of anatase TiO$_2$ is about 3.26 eV), when irradiated by incident light with an energy greater than or equal to its energy gap (E$_g$), the electrons in the valence band absorb photons and are excited, jumping from the valence band to the conduction band, leaving the holes in the valence band, resulting in the formation of the so-called electron (e$^-$)-hole (h$^+$) pairs, the This results in the formation of so-called electron (e$^-$)-hole (h$^+$) pairs, or photogenerated carriers. The electrons (e$^-$) generated by photoexcitation can directly reduce organic matter (such as Dye) or react with electron acceptors; while the holes (h$^+$) generated by photoexcitation can oxidize organic matter or oxidize water and OH$^-$ ions into HO$^\bullet$, and the generated HO$^\bullet$ are very reactive, and can degrade almost all organic matter. This theory qualitatively describes the generation and transfer of photogenerated carriers, but does not give a clear picture of the pollutant degradation process that occurs on the TiO$_2$ surface.

5.  In the manuscript, the discussion on the toxicity of the substances studied is highly relevant and informative. To enhance the readability and accessibility of this crucial information, it might be beneficial to consider listing the detailed toxicity data in the

supplementary materials. This approach would provide those interested with easy access to the comprehensive data.

*Author reply*: Yes. The detailed toxicity data has been added in the supplementary materials.

**Table S8.** The acute and chronic toxicity of Ph, 4-HBA, VL and their products. (mg L$^{-1}$)

| Compounds | LC50(mg/L) fish | LC50(mg/L) daphnid | LC50(mg/L) Green Algae | ChV(mg/L) fish | ChV(mg/L) daphnid | ChV(mg/L) Green Algae |
|---|---|---|---|---|---|---|
| Ph | 27.70 | 9.64 | 2.40 | 2.61 | 0.97 | 4.53 |
| 4-HBA | 17.30 | 17.00 | 9.19 | 3.61 | 0.17 | 3.20 |
| VL | 19.80 | 19.20 | 10.50 | 3.95 | 0.19 | 3.70 |
| P1 | 7.29 | 10.20 | 1.84 | 0.65 | 1.72 | 0.59 |
| P2 | 22.20 | 256.00 | 4.90 | 14.20 | 101.00 | 0.57 |
| P3 | 27.60 | 325.00 | 5.87 | 17.80 | 129.00 | 0.68 |
| P4 | 129.00 | 2.23E+3 | 15.60 | 96.20 | 940.00 | 1.55 |
| P5 | 491.00 | 162.00 | 42.30 | 45.70 | 25.90 | 77.70 |
| P6 | 336.00 | 4420.00 | 75.20 | 238.00 | 1760.00 | 8.57 |
| P7 | 315.00 | 1530.00 | 72.00 | 199.00 | 1400.00 | 8.43 |
| P8 | 3220.00 | 67500.00 | 291.00 | 2590.00 | 2930.00 | 27.00 |
| P9 | 579.00 | 2460.00 | 136.00 | 5430.00 | 869.00 | 1300.00 |
| P10 | 19.20 | 73.00 | 3.45 | 81.30 | 16.70 | 30.70 |
| P11 | 121.00 | 143.00 | 62.40 | 66.20 | 0.96 | 16.00 |

6. The term "Criegee intermediate" in its singular form typically refers to a specific type of intermediate, while the plural form, "Criegee intermediates," refers to multiple types of Criegee intermediates. Therefore, if referring to a particular intermediate, the

singular form should be used; if discussing multiple different intermediates, the plural form should be applied. Line 321, please check "Criegee intermediate".

*Author reply*: Line 409, "Criegee intermediate" has been modified to "Criegee intermediates". Line 298-300, "The ozonolysis of PhCs involves the synthesis of primary ozonide, the formation of active Criegee intermediate (CI), and the disintegration of CI (Rynjah et al., 2024) ".

7. Please give more explanations for atmospheric implications for present results. Have field measurements supported these calculated conclusions?

*Author reply*: The last paragraph in the manuscript states the environmental impact. Line 559:

"Ph undergoes transformation to malealdehyde and catechol when exposed to $O_3$ or $HO^{\bullet}$ in the troposphere (Xu and Wang, 2013). When Ph/VL is at the droplet aerosol interface, rapid oxidation to polyhydroxylated compounds occurs (Ma et al., 2021). VL eventually creates tiny molecule aldehydes and acids. This is consistent with experimental observations (Rana and Guzman, 2020). Li et al. found that seasonal average concentrations of total nitrophenol compounds in particulate matter were comparable to those measured in the gas phase (Li et al., 2022) . However, the reactivity order of nitrophenols in the atmospheric compartments is water droplets > gas phase > particles (Vione et al., 2009). The formation of some low molecular weight acids and aldehydes (2,3-dihydroxymalealdehyde, 2,3-dioxpropanoic acid, etc.) confirms their association with the formation of secondary organic aerosols (SOA). This oxidation process is accelerated when VL is encased in a mineral aerosol represented by $TiO_2$

clusters. It is recommended that enterprises producing lignin, such as those in the pulp and paper industry, or factories that employ lignin in the manufacturing of adhesives, rust inhibitors, color dispersants, diluents, or other similar products, be constructed in regions with low relative humidity. It is recommended that treatment facilities that collect lignin pyrolysis products and recycle the byproducts be located in the surrounding area".

Besides, experimental results have been included for comparison in the Results and Discussion section of the manuscript.

Line 323:"The worldwide mean tropospheric concentration of HO$^\bullet$ is roughly $11.3 \times 10^5$ molecules cm$^{-3}$ (Lelieveld et al., 2016) ".

Line 298: The ozonolysis of PhCs involves the synthesis of primary ozonide, the formation of active Criegee intermediate (CI), and the disintegration of CI (Rynjah et al., 2024) .

Line 375:"This is consistent with previous studies that electron density influences the oxidative activity of PhCs (Rana and Guzman, 2022b) ".

Line 417:"P2 generates o-semiquinone radicals via pathways $R_{HAA}$ by HO$^\bullet$ or $O_3$, which in turn generate oligomers (Guzman et al., 2022). This results in the formation of brown organic carbon in atmospheric aerosols".

Line 426: "These hydroxylation products have been detected by experimental means (Pillar-Little et al., 2014; Pillar-Little and Guzman, 2017; Rana and Guzman, 2020) ".

8.    Some details need to be modified. In line 384, change "k" to italic "$k$". In line 121 currently lacks a leading space. Please insert a space at the beginning of this paragraph

to maintain the uniform indentation level throughout the document. Consistency in formatting contributes to the overall readability and professional appearance of your manuscript. The term "A-Winterface" mentioned in line 240 appears to be a typographical error. Please correct it to "A-W interface" to ensure clarity and correct representation of the term. In line 278, "titanium atom" should be changed to "Ti atom". There is an inconsistency in the format of the thermodynamic symbols in line 337. Specifically, "$\Delta$rG" and "$\mathbf{D}G^{\ddagger}$" use different styles for the delta symbol ($\Delta$). Please standardize the formatting of these symbols to ensure uniformity throughout the document. In line 546, please check "68%". In line 916, do not italicize the subscript of k, such as "$k_{\text{total-A-W, cal}}$".

*Author reply*: In line 124, a leading space has been added.

"Considering the significance of the interfacial behavior of Ph, 4-HBA, and VL at the A-W interface, the properties of these three substances were initially examined by focusing on the A-W interface. **Fig. S1 (a)** depicts a rectangular box that has dimensions of $4 \times 4 \times 9$ nm$^3$ and has a Z-axis that is perpendicular to the A-W contact. This box was used for all simulations. To begin the process of constructing the initial configurations, a water slab measuring $4 \times 4 \times 4$ nm$^3$ was positioned at the coordinates (2 nm, 2 nm, 4.5 nm) of the center of mass (COM). Because the rest extension along the Z-axis of the box was sufficiently large (2.5 nm$^3$), it was possible to steer clear of the intersection of two A-W interface. Prior to the formal simulation, six Ph molecules were randomly placed in a vacuum above the water box for 150 nanoseconds of NVT molecular-dynamics simulations. The results show that there are no significant $\pi$-$\pi$

interactions or formation of hydrogen bonds between the Ph molecules. To simplify the model, this was followed by simulations of individual molecules. Ph, 4-HBA, or VL were each placed in their own compartment at the coordinates (2.0 nm, 2.0 nm, 7.75 nm) for each system in order to simulate the behavior of these molecules in the gas-water interface region of nanobubbles. To begin, the three different systems were optimized to use the least amount of energy possible. After that, NVT molecular-dynamics simulations were carried out for a total of 150 nanoseconds."

In **Text S7**, "A-Winterface" has been modified to "A-W interface".

In **Text S8**, "titanium atom" has been modified to "Ti".

Line 318, "$\mathbf{D}G^{\ddagger}$" has been modified to "The effect of solvation on $\Delta G^{\ddagger}$ is predominantly caused by the hydration of the phenolic OH group, as this is the part of the molecule being dissolved".

Line 980, the title of Table 1 has been centered.

**Reviewer #2:** The article written by Huo et al., is generally well written and present theoretical investigations of the three selected aromatic compounds regarding their degradation mechanism and reactivity initiated by ozone and OH radicals at different environmental media.

Please revise the manuscript according to following comments and advices:

1. The numbers, percentages and any value included in the paper should present a range of uncertainties if the value are coming from experimental investigations.

*Author reply*: Thanks for your suggestions. A range of uncertainties has been given for the experimental data. Line 980:

Table 1 The available experimental and calculated reaction rate constants ($k$) values of $O_3$-initiated and $HO^\bullet$-initiated reactions at 298 K. Unit: $cm^3$ molecule$^{-1}$ s$^{-1}$.

| Compounds | $k_{tot\text{-}A\text{-}W,cal}$[a] | $k_{tot\text{-}TiO_2,cal}$[b] | $k_{tot\text{-}gas,cal}$[c] | $k_{tot\text{-}wat,cal}$[d] | $k_{,exp}$ | Ref. |
|---|---|---|---|---|---|---|
| Ph | $5.98 \times 10^{-7}$ | $1.84 \times 10^{-24}$ | $5.27 \times 10^{-20}$ | $4.02 \times 10^{12}$ | $(13.5 \pm 1.1) \times 10^{-18,}$[e] | Zein et al. (2015) |
|  | $2.69 \times 10^{-6}$ | $3.17 \times 10^{-10}$ | $2.34 \times 10^{-9}$ | $4.46 \times 10^{13}$ | — |  |
| 4-HBA | $6.79 \times 10^{-23}$ | $5.32 \times 10^{-24}$ | $4.93 \times 10^{-24}$ | $1.97 \times 10^{12}$ | — |  |
|  | $9.49 \times 10^{-11}$ | $3.16 \times 10^{-9}$ | $7.90 \times 10^{-11}$ | $2.52 \times 10^{13}$ | — | Rana et al. (2020) |
| VL | $1.27 \times 10^{-17}$ | $3.30 \times 10^{-15}$ | $1.35 \times 10^{-22}$ | $2.20 \times 10^{12}$ | $(0.40 \pm 0.31) \times 10^{-18,}$[f] | Zein et al. (2015) |
|  | $1.73 \times 10^{-7}$ | $6.70 \times 10^{-6}$ | $1.14 \times 10^{-10}$ | $3.15 \times 10^{13}$ | $6.00 \times 10^{-11,}$[g] | Rana et al. (2020) |

[a]: calculated values of phenolic compounds at A-W interface;
[b]: calculated values of phenolic compounds on $TiO_2$ clusters;
[c]: calculated values of phenolic compounds in the gas phase;
[d]: calculated values of phenolic compounds in the bulk water.
[e]: experimental values of catechol in the gas phase;
[f]: experimental values of guaiacol in the gas phase;
[g]: experimental average $k_{HO^\bullet}$ values of methoxyphenols (MPs) in the gas phase.

2. The selection of the model compounds is not enough explained. Increasing number of constituents on the aromatic ring would affect the reactivity and would lead to complex compounds following reaction addition and/or open ring pathway.

*Author reply*: The reason why the model compounds were selected has been added to line 104: "Increasing the number of constituents on the aromatic ring would affect the reactivity and lead to complex compounds after reaction addition and/or open ring pathways".

3. Please follow with attention the way of presenting the interaction of the phenolics with environmental media. The use of subscripts for this should be kept for all of them.

*Author reply*: For this manuscript, these have been modified. Line 38:

"The $O_3$- and $HO^\bullet$-initiated reaction rate constant ($k$) values follow the order of A-W$_{Ph}$ > TiO$_{2\ VL}$ > A-W$_{VL}$ > A-W$_{4-HBA}$ > TiO$_{2\ 4-HBA}$ > TiO$_{2\ Ph}$ and TiO$_{2\ VL}$ > A-W$_{Ph}$ > A-W$_{VL}$ > TiO$_{2\ 4-HBA}$ > TiO$_{2\ Ph}$ > A-W$_{4-HBA}$, respectively".

Line 210: "A-W$_{4-HBA}$".

Line 211: "A-W$_{VL}$".

Line 479: "The following is an order of the $k$ values for $O_3$-initiated reactions: A-W$_{Ph}$ > TiO$_{2\ VL}$ > A-W$_{VL}$ > A-W$_{4-HBA}$ > TiO$_{2\ 4-HBA}$ > TiO$_{2\ Ph}$ (**Fig. 7(a)**). According to **Fig. 7(b)**, the $k$ values of $HO^\bullet$-initiated reactions go as follows: TiO$_{2\ VL}$ > A-W$_{Ph}$ > A-W$_{VL}$ > TiO$_{2\ 4-HBA}$ > TiO$_{2\ Ph}$ > A-W$_{4-HBA}$".

Line 548: " (4) The $k$ values (in molecules·cm$^{-3}$ s$^{-1}$, at 298K and 1 atm) of $O_3$-initiated reactions follow the order of A-W$_{Ph}$ ($5.98 \times 10^{-7}$) > TiO$_{2\ VL}$ ($3.30 \times 10^{-15}$) > A-W$_{VL}$ ($1.27 \times 10^{-17}$) > A-W$_{4-HBA}$ ($6.79 \times 10^{-23}$) > TiO$_{2\ 4-HBA}$ ($5.32 \times 10^{-24}$) > TiO$_{2\ Ph}$ ($1.84 \times 10^{-24}$). The $k$ values of $HO^\bullet$-initiated reactions follow the order of TiO$_{2\ VL}$ ($6.70 \times 10^{-6}$) > A-W$_{Ph}$ ($2.69 \times 10^{-6}$) > A-W$_{VL}$ ($1.73 \times 10^{-7}$) > TiO$_{2\ 4-HBA}$ ($3.16 \times 10^{-9}$) > TiO$_{2\ Ph}$ ($3.17 \times 10^{-10}$) > A-W$_{4-HBA}$ ($9.49 \times 10^{-11}$) ".

4.   Please add minimum information in the article body as the supplementary text would help understanding on main article but not replace it. As an example, a sentence about the kinetic calculation would be necessary in the main text with additional material in SM.

*Author reply*: Some basic computational details have been added to the SM. And it is stated in the body of the text.

Line 157: "details about WHAM are in the Supporting Information **Text S1**".

Line 160: "Estimating the strength of hydrogen bonds (HB) between specific atoms can be done with the help of a tool known as the radial distribution function (RDF). **Text S2** has an explanation of the peculiarities of the RDF and the coordination number".

Line 171: "**Text S3** contains a description of the additional calculated details".

Line 178: "Interaction Region Indicator (IRI) (Lu and Chen, 2021) was used to determine the chemical bonds and weak interactions of Ph/4-HBA/VL adsorbed to $TiO_2$ clusters (the details are in **Text S4**) ".

Line 182: "**Text S5** contains an explanation of the kinetic calculations".

5.   The ozone concentration is known to be variable, however new reference data regarding the more recent measurements of the ozone concentration are necessary.

*Author reply*: The ozone concentration has been updated in the manuscript.

Line 295: "$O_3$ is a major oxidant in the atmosphere, with high concentrations in the troposphere ranging between $9.85 \times 10^{11}$ molecules cm $^{-3}$ (Tomas et al., 2003; Pillar-Little et al., 2014) ".

6.  The gas-phase degradation mechanism initiated by Ozone requires much more attention. Simply Criegee pathway as in the text is not valid, but cleavage of the double bond following a Criegee mechanism from the cyclic adduct would be possible. Please revise the mechanism in details.

*Author reply*:

The ozonolysis mechanism of PhCs is believed to follow the "Criegee mechanism" proposed by Rudolf Criegee (Criegee, 1975). In the classical ozonolysis of unsaturated bonds on olefins (in above Figure), for example, first the unsaturated C=C bond or the unsaturated bond on the benzene ring breaks and cycloadditions onto the 1,3-oxygen atom of the ozone to form an unstable five-membered ring of the primary ozonide (POZ); then the five-membered ring of the POZ is decomposed into a carbonyl compound ($R_1R_2C=O$) and Criegee Intermediates (CIs) ($R_3R_4C=O-O$). All ozonolysis reactions in this paper are designed according to "Criegee mechanism".

**Fig. S11** The subsequent reaction mechanisms of important intermediates (IMs) at A-W interface. Unit in kcal mol $^{-1}$.

7. The concentration of OH radicals is also outdated. Please use more recent information: Lelieveld, J., Gromov, S., Pozzer, A., and Taraborrelli, D.: Global tropospheric hydroxyl distribution, budget and reactivity, Atmos. Chem. Phys., 16, 12477–12493, https://doi.org/10.5194/acp-16-12477-2016, 2016.

*Author reply*: The concentration of OH radicals has been updated in Line 323:

"The worldwide mean tropospheric concentration of HO$^{\bullet}$ is roughly $11.3 \times 10^5$ molecules cm $^{-3}$ (Lelieveld et al., 2016) ".

8. Please add reference in the text every time you add information about products formation, kinetic constants, etc. For example, for catechol formation in gas-phase

*Author reply*: Some references have been added to this manuscript.

Line 559: "Ph undergoes transformation to malealdehyde and catechol when exposed to $O_3$ or $HO^•$ in the troposphere (Xu and Wang, 2013) ".

Line 76: "After being released into the atmosphere, PhCs will be oxidized by ozone ($O_3$) and hydroxyl radicals ($HO^•$). Both are significant contributors to SOA (Arciva et al., 2022) ".

Line 298: "The ozonolysis of PhCs involves the synthesis of primary ozonide, the formation of active Criegee intermediate (CI), and the disintegration of CI (Rynjah et al., 2024) ".

Line 375:"This is consistent with previous studies that electron density influences the oxidative activity of PhCs (Rana and Guzman, 2022b) ".

Line 417:"P2 generates o-semiquinone radicals via pathways $R_{HAA}$ by $HO^•$ or $O_3$, which in turn generate oligomers (Guzman et al., 2022). This results in the formation of brown organic carbon in atmospheric aerosols".

Line 426: "These hydroxylation products have been detected by experimental means (Pillar-Little et al., 2014; Pillar-Little and Guzman, 2017; Rana and Guzman, 2020) ".

9. Please relate the findings more in detail with experimental data.

*Author reply*: Some experimental data has been added to this manuscript. Line 426: "These hydroxylation products have been detected by experimental means (Pillar-Little et al., 2014; Pillar-Little and Guzman, 2017; Rana and Guzman, 2020) ".

Line 566: "Li et al. found that seasonal average concentrations of total nitrophenol compounds in particulate matter were comparable to those measured in the gas phase

(Li et al., 2022) . However, the reactivity order of nitrophenols in the atmospheric compartments is water droplets > gas phase > particles (Vione et al., 2009). The formation of some low molecular weight acids and aldehydes (2,3-dihydroxymalealdehyde, 2,3-dioxpropanoic acid, etc.) confirms their association with the formation of secondary organic aerosols (SOA) ".

Line 503:

"Previous studies measured the second order rate constants of guaiacylacetone + HO$^\bullet$ reaction to be $(14-25) \times 10^9 \, M^{-1} \, s^{-1}$ at pH 5 and 6 at aqueous secondary organic aerosol, which is lower than our results (Arciva et al., 2022). This is because galactose reduces the steady-state concentration of HO$^\bullet$".

Line 470:

"**3.4 Comparison with available experimental results**

The rate constants ($k$) of the overall reaction under the temperature range of 278–318 K were computed based on acquired potential energy surfaces for the O$_3$-initiated and HO$^\bullet$-initiated reactions of selected compounds. The results of these calculations are listed in **Table S5** and **S6**, respectively. The temperature dependences of the various $k$ values for Ph, 4-HBA, and VL at the A-W interface and in bulk water are depicted in **Fig. 7**. At low values of $k$, there is a positive dependence on temperature. When the $k$ values are raised to a certain degree, the temperature dependency seems to lose any significance it may have had before. The following is an order of the $k$ values for O$_3$-initiated reactions: A-W$_{Ph}$ > TiO$_2$ $_{VL}$> A-W$_{VL}$ > A-W$_{4\text{-}HBA}$ > TiO$_2$ $_{4\text{-}HBA}$ > TiO$_2$ $_{Ph}$ (**Fig. 7(a)**). According to **Fig. 7(b)**, the $k$ values of HO$^\bullet$-initiated reactions go as follows: TiO$_2$

$_{VL}$ > A-W $_{Ph}$ > A-W$_{VL}$> TiO$_2$ $_{4-HBA}$ > TiO$_2$ $_{Ph}$> A-W$_{4-HBA}$. In **Fig. 7(a)** and **Fig. 7(b),** the

$k$ values of HO$^{\bullet}$-initiated reactions are one hundred times greater than those of O$_3$-

initiated reactions. **Table 1** is a listing of the experimental and estimated $k$ values that

are available for O$_3$-initiated and HO$^{\bullet}$-initiated reactions at 298 K. According to the

findings, the ozonolysis of Ph was promoted by the water-gas interface as well as by

TiO$_2$ clusters, and the HO$^{\bullet}$ initiated reactions of VL were promoted by TiO$_2$ clusters.

However, the O$_3$/HO$^{\bullet}$ + 4-HBA reactions have the lowest $k$ values among the three

molecules when tested in a variety of environmental environments. The estimated

$k_{O_3+Ph}$ values at the A-W interface are 11 orders of magnitude greater than those of

catechol under dry conditions in gas phase (Zein et al., 2015), when compared with the

experimental data. Because it has a higher $k_{O_3}$ value, catechol, which is one of the main

products of Ph's oxidation in the atmosphere, has a higher degree of reactivity than its

parent compound (**Table 1**). The estimated value of VL is lower than the experimentally

determined value of $k_{O_3}$ for guaiacol under dry conditions, which is $(0.40 \pm 0.31) \times$

$10^{-18}$ cm$^3$ molecule$^{-1}$ s$^{-1}$ in the gas phase (Zein et al., 2015). The difference between

the predicted value of $k_{HO^{\bullet}+VL}$ is $1.14 \times 10^{-10}$ cm$^3$ molecule$^{-1}$ s$^{-1}$ and the average

experimental value of $k_{HO^{\bullet}}$ for methoxyphenols is just an order of magnitude. As a

consequence, the findings of our calculations are reliable.

10. In the environments where the phenolic compounds are released, the presence of

NOx is well represented. Please describe the mechanistic implication of the NOx

system in the degradation product formation. Discuss about the nitroaromatic formation

and their atmospheric behavior".

*Author reply*: That part has been added to the manuscript. Line 426:

"These hydroxylation products have been detected by experimental means (Pillar-Little et al., 2014; Pillar-Little and Guzman, 2017; Rana and Guzman, 2020). The HO• abstracts a hydrogen atom from the hydroxyl group of catechol, forming $C_6H_5O_2$ radical and a water molecule. Due to the widespread presence of $NO_2$ in the environment, it adds to the $C_6H_5O_2$ radical at the ortho position of the extracted hydrogen atom through an addition reaction. Subsequently, a hydrogen transfer reaction occurs, resulting in the formation of 4-nitrobenzene-1,2-diol. This computational result validates the previous experimental hypothesis by Finewax et al (Finewax et al., 2018). 4-Nitrobenzene-1,2-diol subsequently transform into benzoquinone, maleic acid, fumaric acid, acetic anhydride, acetic acid, and formic acid, or are directly mineralized into carbon dioxide and water (Chen et al., 2015) ".

**Fig.6** (Continue) Subsequent reaction mechanisms of important intermediates (IMs)

(unit: kcal mol$^{-1}$) in **(a)** gas phase (g) / bulk water (l) and at **(b)** A-W interface.

11. The article considers multiple oxidation and final formation of benzene hexaol. Is this feasible energetically? Could be formed in the gas-phase?

*Author reply*: It has been added to this manuscript. Line 420:

"At the A-W interface, a sequence of hydroxylation products, including pyrocatechol (P2), benzene-1,2,3-triol (P3), and benzene-1,2,3,4,5-pentaol (P4), are generated through hydroxylation processes rather than by a single SET ($\Delta G^{\ddagger} = 111.79$ kcal mol $^{-1}$). It is difficult for P4 to form benzene-1,2,3,4,5,6-hexaol because hydrogen transfer reactions are difficult to occur ($\Delta G^{\ddagger} = 34.32$ kcal mol $^{-1}$) ". It could not be formed in the gas phase because it is easier to take place at the A-W interface in the presence of water molecules. However, it could not be formed at the A-W interface.

**Fig.6** Subsequent reaction mechanisms of important intermediates (IMs) (unit: kcal mol

$^{-1}$) in **(a)** gas phase (g) / bulk water (l) and at **(b)** A-W interface.

12. Please add information regarding the equilibrium of the gas-particle composition of the three selected aromatics and the importance of them for secondary organic aerosol formation.

*Author reply*: Line 556:

"Li et al. found that seasonal average concentrations of total nitrophenol compounds in particulate matter were comparable to those measured in the gas phase (Li et al., 2022) . However, the reactivity order of nitrophenols in the atmospheric compartments is water droplets > gas phase > particles (Vione et al., 2009). The formation of some low molecular weight acids and aldehydes (2,3-dihydroxymalealdehyde, 2,3-dioxpropanoic acid, etc.) confirms their association with the formation of secondary organic aerosols (SOA) ". The findings underscore the importance of phenol, 4-hydroxybenzaldehyde, and vanillin in urban air quality and highlight the need for targeted emission control strategies to mitigate SOA formation.

13. Please present the degradation mechanism for the reaction of vanillin too.

The degradation mechanism for the reaction of vanillin has been added to Line 462.

"The VL subsequent reaction mechanism is demonstrated in **Fig. S11(d)**. The final oxidation products of VL are P12 ((2E,4E)-4-formyl-2-methoxy-6-oxohexa-2,4-dienoic acid), P13 (ethene-1,1,2-tricarbaldehyde), P14 (oxalaldehyde) and P15 ((E)-2-methoxy-4,5-dioxopent-2-enoic acid). The formation of these products could explain the biomass burning material for the formation of SOA (Rana and Guzman, 2022c) ".

(d)

[Figure]

**Fig. S11** The subsequent reaction mechanisms of important intermediates (IMs) at A-W interface. Unit in kcal mol $^{-1}$.

Line 281 – "phosphorus" ?

Author reply: Line 258, the "phosphorus atoms" has been modified to "Ph".

Line 349 – Authors are mentioning the ozone reaction. Is that true?

Author reply: It has been modified to "HO$^{\bullet}$ + PhCs reactions".

Line 348-350 – Please revise the sentence since there is a mix between OH and O3 reactions.

Author reply: Line 325, it has been modified to "For this reason, elucidating the reaction mechanism underlying HO$^{\bullet}$ + PhCs reactions in the troposphere is of the utmost importance".

Line 380 – "Phe + O3 reactions"

Author reply: Thank you for your feedback, but upon careful review, I believe the original text is correct as written.

**Reviewer #3:**

**General Comments:** The manuscript uses molecular dynamics simulation and quantum chemical calculations to explore the effect of selected environments on the initial reaction of three phenols (phenol, vanillin, and 4-hydroxybenzaldehyde) with ozone and OH radical. There are multiple concerns listed in the major comments that will require extensive work for the manuscript to be considered for publication in ACP. In its current form, the manuscript cannot be recommended for publication.

**Major Comments:**

1) The manuscript that does not seem to fit the "Measurement report" type expected for the peer review in Atmospheric Chemistry and Physics (ACP). Such type of manuscript should present substantial new results from measurements of atmospheric properties and processes from field and laboratory experiments. There are no field observations or laboratory experiment measurements in this manuscript that could be modeled to gain new insights to advance the field.

*Author reply*: Thank you very much for your thorough review of our manuscript. I greatly appreciate the insightful comments and suggestions you provided. I have found them to be highly beneficial and have learned a lot from them. Thank you for your insightful feedback regarding the classification of our manuscript as a "Measurement report." I would like to note that the decision to classify the manuscript under this category was based on specific suggestions from the editor. Given the apparent discrepancy between the editor's instructions and the journal's expectations as you have outlined, I would appreciate further guidance. I have reviewed the manuscript types for

the ACP journal available at https://www.atmospheric-chemistry-and-physics.net/about/manuscript_types.html. I will consult with the editor again regarding this modification.

2) The covered manuscript theme of phenols' reactions with $O_3$ and OH radical at the air-water interface has been extensively studied in the literature. However, such material is poorly covered in the manuscript, with the relevant literature missing from the bibliography. The resulting manuscript provided is a very incomplete account of relevant works of phenols' reactions with $O_3$ and OH radical at the air-water interface, in water, and in the gas phase. The manuscript did not appropriately connect the introduction, discussion, and conclusions sections to the published peer-review literature.

*Author reply*: Some references has been added to this manuscript.

Line 76:

"After being released into the atmosphere, PhCs will be oxidized by ozone ($O_3$) and hydroxyl radicals (HO$^\bullet$). Both are significant contributors to SOA (Arciva et al., 2022) ".

Line 83:

"Furthermore, they investigated the hydroxylation, ring opening, and oligomerization processes of PhCs in the atmospheric liquid phase, with a focus on the potential environmental toxicity and climatic effects of these events (Ma et al., 2021; Liu et al., 2022a; Arciva et al., 2022; Carena et al., 2023) ".

Line 90:

"The atmosphere contains a high concentration of aqueous aerosols and water microdroplets (Zhong et al., 2019; Guzman et al., 2022). The oxidation of PhCs can rapidly occur at A-W interface (Rana and Guzman, 2022a) ".

Line 76: "After being released into the atmosphere, PhCs will be oxidized by ozone ($O_3$) and hydroxyl radicals ($HO^\bullet$). Both are significant contributors to SOA (Arciva et al., 2022) ".

Line 294: "$O_3$ is a major oxidant in the atmosphere, with high concentrations in the troposphere ranging between $9.85 \times 10^{11}$ molecules cm $^{-3}$ (Tomas et al., 2003; Pillar-Little et al., 2014) Investigating the fate of PhCs in the presence of $O_3$ is essential (Pillar-Little et al., 2014; Rana and Guzman, 2020) ".

Line 323:"The worldwide mean tropospheric concentration of $HO^\bullet$ is roughly $11.3 \times 10^5$ molecules cm $^{-3}$ (Lelieveld et al., 2016) ".

Line 298: The ozonolysis of PhCs involves the synthesis of primary ozonide, the formation of active Criegee intermediate (CI), and the disintegration of CI (Rynjah et al., 2024) .

Line 375: "This is consistent with previous studies that electron density influences the oxidative activity of PhCs (Rana and Guzman, 2022b) ".

Line 417: "P2 generates o-semiquinone radicals via pathways $R_{HAA}$ by $HO^\bullet$ or $O_3$, which in turn generate oligomers (Guzman et al., 2022). This results in the formation of brown organic carbon in atmospheric aerosols".

Line 426: "These hydroxylation products have been detected by experimental means (Pillar-Little et al., 2014; Pillar-Little and Guzman, 2017; Rana and Guzman, 2020) ".

Line 503:

"Previous studies measured the second order rate constants of guaiacylacetone + HO•

reaction to be $(14-25) \times 10^9\,M^{-1}\,s^{-1}$ at pH 5 and 6 at aqueous secondary organic aerosol,

which is lower than our results (Arciva et al., 2022). This is because galactose reduces

the steady-state concentration of HO•. The reaction rate constants of PhCs increase with

increasing pH and we calculated the rate constants at pH 7 in bulk water (Ma et al.,

2021) ".

3) The abstract needs to be completely rewritten for clarity as detailed in the next points

4 through 7.

*Author reply*: Thank you for your suggestions. The abstract has been rewritten

according your suggestions as detailed in the next points 4 through 7.

4) l. 24-30: The text did not acknowledge that the mechanisms and kinetics for the

reactions of phenol, vanillin, and 4-hydroxybenzaldehyde at the air-water interface, in

the gas-phase, and in bulk water have been studied experimentally in detail recently.

Therefore, what is the novelty of performing molecular dynamics simulations and

quantum chemical calculations here?

*Author reply*: Although mechanisms and kinetics for the reactions of phenol, vanillin,

and 4-hydroxybenzaldehyde at the air-water interface, in the gas-phase, and in bulk

water have been studied experimentally, no comparisons have been made of the effects

of different environmental media on the oxidative activity of these compounds.

Experimentally, some reaction mechanisms can be deduced from the measured

products. However, theoretical calculations show quantitatively whether the reaction

mechanism occurs, and if not, give possible reaction paths by theoretical calculations. Due to the word limit of the abstract, the following changes have been made to line 26: "To address the gaps in experimental research, phenol (Ph), 4-hydroxybenzaldehyde (4-HBA), and vanillin (VL) are chosen as model compounds to investigate their reaction mechanism and kinetics at the air-water (A-W) interface, on $TiO_2$ clusters, in the gas phase, and in bulk water using a combination of molecular dynamics simulation and quantum chemical calculations".

5) l. 31-33: It is completely unclear where the said occurrence percentages of the 3 phenols at the air-water interface come from and their validity. What is a $(TiO_2)_n$ cluster and why should the reader of ACP care about it? What do the authors mean by "adsorption capacity" and "the capacity" and why is it relevant here? The text lacks focus and clarity.

*Author reply*: Occurrence percentages of the 3 phenols at the air-water interface have been explained in line $186 \sim 193$ in the manuscript: "According to location definitions, $O_3$ percentage distribution was as follows: 26% at the A-W interface; 72% in the air; and 2% in pure water (**Fig. 1(b)**). **Fig. 1(c)** depicts MD trajectories of Ph diffusion through the water slab from the air region over a 150 ns period. Ph is distributed in the air (8%) and bulk water (20%), with the majority at the A-W interface (72%) (**Fig. 1 (d)**). The majority of 4-HBA and VL molecules are located at the A-W interface, constituting 68% and 73% of the total locations as presented in **Fig. S2**. " It is clear from the context that they occur in the tropospheric atmosphere. Line $24 \sim 26$, "Environmental media affect the atmospheric oxidation processes of phenolic

compounds (PhCs) released from biomass burning in the troposphere". Given the length constraints of abstract, this is not stated again here.

The reason why should the reader of ACP care about it has been explained in line 95 ~ 96:

"In chemical engineering, titanium dioxide ($TiO_2$) is an essential photoactive component found in atmospheric mineral dust (Sakata et al., 2021; Wang et al., 2023). The interaction between PhCs and $TiO_2$ is continuous (Grassian, 2009; Rubasinghege et al., 2010; Shang et al., 2021), despite the relatively low prevalence of $TiO_2$ mineral particles (comprising 0.1% to 10% by mass). Therefore, it is essential to investigate the disparity in the oxidation reaction mechanisms and kinetics of PhCs at A-W interface and mineral dust particles".

Line 29, "on $TiO_2$ clusters" has been modified to "on $TiO_2$ mineral aerosols".

The "adsorption capacity" is one technical term (Liu et al., 2021; Liu et al., 2022b; Zhang et al., 2023). In this manuscript, adsorption capacity refers to the magnitude of the adsorption energy.

6) l. 36-38: What environmental media do the authors refer to? None is distinguished in the statement. What specific rate constants for $O_3$ and OH radical initiated reactions (of the many possible based on the literature but not covered in the manuscript) are compared? The provided comparison of the presented calculations without a connection to the scientific literature is of no value to advance this field.

*Author reply*: Environmental media has been written in line 28: "air-water (A-W) interface, on $TiO_2$ mineral aerosols, in the gas phase, and in bulk water".

Line 470:

"**3.4 Comparison with available experimental results**

The rate constants ($k$) of the overall reaction under the temperature range of 278–318 K were computed based on acquired potential energy surfaces for the $O_3$-initiated and $HO^•$-initiated reactions of selected compounds. The results of these calculations are listed in **Table S5** and **S6**, respectively. The temperature dependences of the various $k$ values for Ph, 4-HBA, and VL at the A-W interface and in bulk water are depicted in **Fig. 7**. At low values of $k$, there is a positive dependence on temperature. When the $k$ values are raised to a certain degree, the temperature dependency seems to lose any significance it may have had before. The following is an order of the $k$ values for $O_3$-initiated reactions: A-W$_{Ph}$ > TiO$_2$ $_{VL}$> A-W$_{VL}$ > A-W$_{4-HBA}$ > TiO$_2$ $_{4-HBA}$ > TiO$_2$ $_{Ph}$ (**Fig. 7(a)**). According to **Fig. 7(b)**, the $k$ values of $HO^•$-initiated reactions go as follows: TiO$_2$ $_{VL}$ > A-W $_{Ph}$ > A-W$_{VL}$> TiO$_2$ $_{4-HBA}$ > TiO$_2$ $_{Ph}$> A-W$_{4-HBA}$. In **Fig. 7(a)** and **Fig. 7(b)**, the $k$ values of $HO^•$-initiated reactions are one hundred times greater than those of $O_3$-initiated reactions. **Table 1** is a listing of the experimental and estimated $k$ values that are available for $O_3$-initiated and $HO^•$-initiated reactions at 298 K. According to the findings, the ozonolysis of Ph was promoted by the water-gas interface as well as by TiO$_2$ clusters, and the $HO^•$ initiated reactions of VL were promoted by TiO$_2$ clusters. However, the $O_3/HO^•$ + 4-HBA reactions have the lowest $k$ values among the three molecules when tested in a variety of environmental environments. The estimated $k_{O_3+Ph}$ values at the A-W interface are 11 orders of magnitude greater than those of catechol under dry conditions in gas phase (Zein et al., 2015), when compared with the

experimental data. Because it has a higher $k_{O_3}$ value, catechol, which is one of the main products of Ph's oxidation in the atmosphere, has a higher degree of reactivity than its parent compound (**Table 1**). The estimated value of VL is lower than the experimentally determined value of $k_{O_3}$ for guaiacol under dry conditions, which is $(0.40 \pm 0.31) \times 10^{-18}$ cm$^3$ molecule $^{-1}$ s $^{-1}$ in the gas phase (Zein et al., 2015). The difference between the predicted value of $k_{HO^\bullet+VL}$ is $1.14 \times 10^{-10}$ cm$^3$ molecule $^{-1}$ s $^{-1}$ and the average experimental value of $k_{HO^\bullet}$ for methoxyphenols is just an order of magnitude. As a consequence, the findings of our calculations are reliable. Previous studies measured the second order rate constants of guaiacylacetone + HO$^\bullet$ reaction to be $(14-25) \times 10^9$ M$^{-1}$ s$^{-1}$ at pH 5 and 6 at aqueous secondary organic aerosol, which is lower than our results (Arciva et al., 2022). This is because galactose reduces the steady-state concentration of HO$^\bullet$. The reaction rate constants of PhCs increase with increasing pH and we calculated the rate constants at pH 7 in bulk water (Ma et al., 2021). This study summarizes the O$_3$- and HO$^\bullet$-initiated reaction sequences of three PhCs in different environmental media. The reaction sequences for O$_3$- and HO$^\bullet$-initiated reactions of Ph and 4-HBA are identical in different environmental media, while VL shows slight variations. For O$_3$-initiated reactions, the reaction sequences are as follows: Ph: A-W interface > Bulk water > Gas phase > TiO$_2$ clusters; 4-HBA: Bulk water > A-W interface > TiO$_2$ clusters > Gas phase; VL: Bulk water > TiO$_2$ clusters > A-W interface > Gas phase. For HO$^\bullet$-initiated reactions, the sequences are: Ph: A-W interface ≈ Bulk water > Gas phase > TiO$_2$ clusters; 4-HBA: Bulk water > A-W interface > TiO$_2$ clusters > Gas phase; VL: TiO$_2$ clusters > Bulk water > A-W interface > Gas phase".

7) l. 40-46: The text should have first defined the products or byproducts that could be more harmful than their parent compounds. A better comparison among media would be beneficial here.

*Author reply*: The toxicity of a compound is closely related to its structure, not the environmental medium. And we are more concerned here with the difference between the toxicity of the product and that of the parent compound. Different environmental media can accelerate or slow down the production of these products. Therefore, we focus on the comparison of reaction rate constants of selected compounds in different environmental media.

8) l. 50-110 (Introduction): The section needs to be completely rewritten and incorporate many key missing papers in the topic including those form searching in the Web of Science and SciFinder-n of key terms that is presented below. The twenty-six papers[1-26] presented below in revere chronological order are not all missing works but just serve as an example of selected literature that needs to be covered in this manuscript:

1. Zhang, J.; Shrivastava, M.; Ma, L.; Jiang, W. Q.; Anastasio, C.; Zhang, Q.; Zelenyuk, A., Modeling Novel Aqueous Particle and Cloud Chemistry Processes of Biomass Burning Phenols and Their Potential to Form Secondary Organic Aerosols. *Environmental Science & Technology* **2024**, *58*, (8), 3776-3786.

2. Rana, M. S.; Bradley, S. T.; Guzman, M. I., Conversion of Catechol to 4-Nitrocatechol in Aqueous Microdroplets Exposed to $O_3$ and $NO_2$. *ACS ES&T Air* **2024,** *1*, (2), 80-91.

3. Ma, L.; Worland, R.; Heinlein, L.; Guzman, C.; Jiang, W. Q.; Niedek, C.; Bein, K. J.; Zhang, Q.; Anastasio, C., Seasonal variations in photooxidant formation and light absorption in aqueous extracts of ambient particles. *Atmospheric Chemistry and Physics* **2024,** *24*, (1), 1-21.

4. Ma, L.; Worland, R.; Jiang, W. Q.; Niedek, C.; Guzman, C.; Bein, K. J.; Zhang, Q.; Anastasio, C., Predicting photooxidant concentrations in aerosol liquid water based on laboratory extracts of ambient particles. *Atmospheric Chemistry and Physics* **2023,** *23*, (15), 8805-8821.

5. Jiang, W. Q.; Niedek, C.; Anastasio, C.; Zhang, Q., Photoaging of phenolic secondary organic aerosol in the aqueous phase: evolution of chemical and optical properties and effects of oxidants. *Atmospheric Chemistry and Physics* **2023,** *23*, (12), 7103-7120.

6. Rana, M. S.; Guzman, M. I., Surface Oxidation of Phenolic Aldehydes: Fragmentation, Functionalization, and Coupling Reactions. *Journal of Physical Chemistry A* **2022,** *126*, (37), 6502-6516.

7. Rana, M. S.; Guzman, M. I., Oxidation of Catechols at the Air-Water Interface by Nitrate Radicals. *Environmental Science & Technology* **2022,** *56*, (22), 15437-15448.

8. Rana, M. S.; Guzman, M. I., Oxidation of Phenolic Aldehydes by Ozone and Hydroxyl Radicals at the Air-Solid Interface. *ACS Earth and Space Chemistry* **2022,** *6*, (12), 2900-2909.

9. Guzman, M. I.; Pillar-Little, E. A.; Eugene, A. J., Interfacial Oxidative Oligomerization of Catechol. *ACS Omega* **2022**.

10. Arciva, S.; Niedek, C.; Mavis, C.; Yoon, M.; Sanchez, M. E.; Zhang, Q.; Anastasio, C., Aqueous middot OH Oxidation of Highly Substituted Phenols as a Source of Secondary Organic Aerosol. *Environmental Science & Technology* **2022,** *56*, (14), 9959-9967.

11. Al-Abadleh, H. A.; Motaghedi, F.; Mohammed, W.; Rana, M. S.; Malek, K. A.; Rastogi, D.; Asa-Awuku, A. A.; Guzman, M. I., Reactivity of aminophenols in forming nitrogen-containing brown carbon from iron-catalyzed reactions. *Communications Chemistry* **2022,** *5*, (1).

12. Ma, L.; Guzman, C.; Niedek, C.; Tran, T.; Zhang, Q.; Anastasio, C., Kinetics and Mass Yields of Aqueous Secondary Organic Aerosol from Highly Substituted Phenols Reacting with a Triplet Excited State. *Environmental Science & Technology* **2021,** *55*, (9), 5772-5781.

13. Rana, M. S.; Guzman, M. I., Oxidation of Phenolic Aldehydes by Ozone and Hydroxyl Radicals at the Air-Water Interface. *Journal of Physical Chemistry A* **2020,** *124*, (42), 8822-8833.

14. Pillar-Little, E. A.; Guzman, M. I., An Overview of Dynamic Heterogeneous Oxidations in the Troposphere. *Environments* **2018,** *5*, (9).

15. Kaur, R.; Anastasio, C., First Measurements of Organic Triplet Excited States in Atmospheric Waters. *Environmental Science & Technology* **2018,** *52*, (9), 5218-5226.

16. Jurak, M.; Mroczka, R.; Lopucki, R., Properties of Artificial Phospholipid Membranes Containing Lauryl Gallate or Cholesterol. *Journal of Membrane Biology* **2018,** *251*, (2), 277-294.

17. Huang, D. D.; Zhang, Q.; Cheung, H. H. Y.; Yu, L.; Zhou, S.; Anastasio, C.; Smith, J. D.; Chan, C. K., Formation and Evolution of aqSOA from Aqueous-Phase Reactions of Phenolic Carbonyls: Comparison between Ammonium Sulfate and Ammonium Nitrate Solutions. *Environmental Science & Technology* **2018,** *52*, (16), 9215-9224.

18. Pillar, E. A.; Guzman, M. I., Oxidation of Substituted Catechols at the Air-Water Interface: Production of Carboxylic Acids, Quinones, and Polyphenols. *Environmental Science & Technology* **2017,** *51*, (9), 4951-4959.

19. Lin, P. C.; Wu, Z. H.; Chen, M. S.; Li, Y. L.; Chen, W. R.; Huang, T. P.; Lee, Y. Y.; Wang, C. C., Interfacial Solvation and Surface pH of Phenol and Dihydroxybenzene Aqueous Nanoaerosols Unveiled by Aerosol VUV Photoelectron Spectroscopy. *Journal of Physical Chemistry B* **2017,** *121*, (5), 1054-1067.

20. Kaur, R.; Anastasio, C., Light absorption and the photoformation of hydroxyl radical and singlet oxygen in fog waters. *Atmospheric Environment* **2017,** *164*, 387-397.

21. Yu, L.; Smith, J.; Laskin, A.; George, K. M.; Anastasio, C.; Laskin, J.; Dillner, A. M.; Zhang, Q., Molecular transformations of phenolic SOA during photochemical

aging in the aqueous phase: competition among oligomerization, functionalization, and fragmentation. *Atmospheric Chemistry and Physics* **2016,** *16*, (7), 4511-4527.

22. Pillar, E. A.; Zhou, R. X.; Guzman, M. I., Heterogeneous Oxidation of Catechol. *Journal of Physical Chemistry A* **2015,** *119*, (41), 10349-10359.

23. Chen, C. M.; Chen, H. S.; Yu, J.; Han, C.; Yan, G. X.; Guo, S. H., *p*-Nitrophenol Removal by Bauxite Ore Assisted Ozonation and its Catalytic Potential. *Clean-Soil Air Water* **2015,** *43*, (7), 1010-1017.

24. Smith, J. D.; Sio, V.; Yu, L.; Zhang, Q.; Anastasio, C., Secondary Organic Aerosol Production from Aqueous Reactions of Atmospheric Phenols with an Organic Triplet Excited State. *Environmental Science & Technology* **2014,** *48*, (2), 1049-1057.

25. Pillar, E. A.; Camm, R. C.; Guzman, M. I., Catechol Oxidation by Ozone and Hydroxyl Radicals at the Air-Water Interface. *Environmental Science & Technology* **2014,** *48*, (24), 14352-14360.

26. M'Hemdi, A.; Dbira, B.; Abdelhedi, R.; Brillas, E.; Ammar, S., Mineralization of Catechol by Fenton and Photo-Fenton Processes. *Clean-Soil Air Water* **2012,** *40*, (8), 878-885.

Many of the above papers should also be recalled in the results and discussion and conclusions sections.

*Author reply*: Thank you for your suggestions. Many of the above papers have been added in the introduction, results and discussion and conclusions sections.

Line 76: "After being released into the atmosphere, PhCs will be oxidized by ozone ($O_3$) and hydroxyl radicals ($HO^{\bullet}$). Both are significant contributors to SOA (Arciva et al., 2022) ".

Line 294: $O_3$ is a major oxidant in the atmosphere, with high concentrations in the troposphere ranging between $9.85 \times 10^{11}$ molecules cm $^{-3}$ (Tomas et al., 2003; Pillar-Little et al., 2014) . Investigating the fate of PhCs in the presence of $O_3$ is essential (Pillar-Little et al., 2014; Rana and Guzman, 2020) ".

Line 323: "The worldwide mean tropospheric concentration of $HO^{\bullet}$ is roughly $11.3 \times 10^5$ molecules cm $^{-3}$ (Lelieveld et al., 2016) ".

Line 298: The ozonolysis of PhCs involves the synthesis of primary ozonide, the formation of active Criegee intermediate (CI), and the disintegration of CI (Rynjah et al., 2024) .

Line 375: "This is consistent with previous studies that electron density influences the oxidative activity of PhCs (Rana and Guzman, 2022b) ".

Line 417: "P2 generates o-semiquinone radicals via pathways $R_{HAA}$ by $HO^{\bullet}$ or $O_3$, which in turn generate oligomers (Guzman et al., 2022). This results in the formation of brown organic carbon in atmospheric aerosols".

Line 426: "These hydroxylation products have been detected by experimental means (Pillar-Little et al., 2014; Pillar-Little and Guzman, 2017; Rana and Guzman, 2020) ".

9) l. 107: This was not an experimental measurement but a quantum chemical calculation. It seems incorrect to state that "Rate constants were measured …".

*Author reply*: Yes, thank you for your suggestions. Line 110: this sentence has been modified to "Rate constants were calculated throughout a wide temperature range in various EM".

10) l. 109-110, l. 502-524: The computational work of toxicological relevance is not of primary interest to the readers of ACP. Such work is out of scope for this journal and should be moved to the accompanying Supplementary Material document.

*Author reply*: Thank you for your suggestions. Line 5523-line 545 has been moved to Supplementary Material document. It has been added to Line 523:

" See Text S9 for ecotoxicity assessment".

11) l. 124: A better contextual explanation to the selected dimensions is needed before the values of $4 \times 4 \times 9$ nm$^3$ are provided.

*Author reply*: The explanation to the selected dimensions has been added to Line 129:

"A water box that is too small may cause the central PhCs molecules to be too close to the interface region, leading to inaccurate results. Conversely, opting for a water box that is too large can lead to unnecessary waste of computational resources".

12) l. 128-130: The statement is unclear for the general reader to understand its intent. The authors should more clearly state what the meaning of "… steer clear of the intersection of two A-W interface" is.

*Author reply*: In line 134 it has been written that the water box is depicted in **Fig. S1 (a)**, where the air-water interface (upper and lower) can be clearly seen.

[Figure]

**Fig. S1** Schematic representation the initial configuration of **(a)** the umbrella sampling simulation and **(b)** the 150 ns NVT simulation (unit: nm).

13) l. 130-131: The manuscript should state how the random selection was executed in this work.

*Author reply*: Random selection is the random selection of one or a few frames inside a simulation of many frames.

14) l. 132: The manuscript should explain to the readers of ACP why such a short 150 ns period is relevant. The explanation should be followed by establishing the relevance of this period to what happens in the environment.

*Author reply*: In fact, the shorter the simulation time, the closer it is to the real situation. The longer the simulation time, the larger the error and the greater the deviation from the real situation. Moreover, we quickly observed PhCs staying at the water-air interface within 150 ns, so there is no need to increase the simulation time.

15) l. 133-134: Clearly state the pi-pi and H-bond molecular orbitals and/or atomic centers that are considered for these interactions of phenols.

*Author reply*: When talking to other researchers, they are concerned about the interactions between phenols at the water-air interface. And the physical interactions between them are mainly considered pi-pi and H-bond molecular orbitals and/or atomic centres, but here we do not observe that they aggregate.

16) l. 137-138: The text appears disconnected to the aimed description and the actual environment. There are no relevant nanobubbles in air to assume such a model of the air-water interface is of any atmospheric relevance.

*Author reply*: Yes, the expression "nanobubbles" is not accurate enough. In fact, this study is based on the cloud/fog drops and aerosol liquid water (ALW) in the atmosphere. It has been modified to "cloud/fog drops and aerosol liquid water (ALW) " in line 145.

17) l. 151 and l. 155: Indicate the "specifics" that can be found in the Supplementary Materials by completing the ideas in these statements. Similarly, in l. 174, expand the text to inform the readers of ACP what calculations are in the Supplementary Materials.

*Author reply*: This section has been modified in the manuscript to make the information complete. Line 154:

"The weighted histogram analysis approach, also known as WHAM, can be used to calculate the free energy profiles of Ph, 4-HBA, or VL when they transition from the gas phase into bulk water (Kumar et al., 1992; Hub et al., 2010) ; details are in the Supporting Information **Text S1**".

Line 162:

"**Text S2** has an explanation of the peculiarities of the RDF and the coordination number".

Line 182:

"**Text S5** contains an explanation of the kinetic calculation methods".

18) l. 157: What electrical structure do the authors refer to? Explain in the text.

*Author reply*: This part has been revised in the manuscript. Line 165:

"In this work, all structural optimization and energy calculation were accomplished by utilizing the Gaussian16 program (Frisch et al., 2016) ".

19) l. 159: What is the meaning of "benchmarking" here? Clarify the text.

*Author reply*: The "benchmark" is a term used in the field of computational chemistry. By comparing different calculation methods, the most suitable calculation method for this study was found. In addition, this sentence also explains in detail the computational methods used by benchmark. Line 163-167:

"By benchmarking at the CCSD(T)/cc-pVDZ, CBS-QB3, B3LYP/6-311+G(d,p), MP2/6-311+G(d,p) and M06-2X/6-311+G(d,p) levels, Cao et al. (Cao et al., 2021) found that M06-2X/6-311++G(3df,2p)//M06-2X/6-31+G(d,p) is reliable for PhCs".

20) l. 162: In what sense and context is this "reliable" for phenols?

*Author reply*: Reaction conditions have been added to the manuscript. Line 167:

"By benchmarking at the CCSD(T)/cc-pVDZ, CBS-QB3, B3LYP/6-311+G(d,p), MP2/6-311+G(d,p) and M06-2X/6-311+G(d,p) levels, Cao et al. (Cao et al., 2021) found that M06-2X/6-311++G(3df,2p)//M06-2X/6-31+G(d,p) is reliable for PhCs at gas phase".

21) l. 163: Several levels were presented above so what is "this level" for the gas phase reactions?

*Author reply*: Based on the context, "this level" refers to "M06-2X/6-311++G(3df,2p)//M06-2X/6-31+G(d,p) ".

22) l 185-185: Range of what? Complete the idea in the statement and preferably give the values in this range.

*Author reply*: In context, the exact range of values is given in the previous sentence. Line 190:

"**Fig. 1(a)** displays the variation in water density along the Z-coordinate distance from 0 to 9 nm, categorizing three zones: A-W interface (2.25 to 2.79 nm and 6.21 to 6.75 nm), air (0 to 2.25 nm and 6.75 to 9 nm), and bulk water (2.79 to 6.21 nm). This method accurately determines the interfacial range (Zhang et al., 2019; Shi et al., 2020) ".

23) l. 188: The concept of "pure" water is completely wrong here.

*Author reply*: Line 196: The "pure water" has been modified to "bulk water".

24) l. 190-193: Explain why most of the phenols positioned themselves at the air-water interface.

*Author reply*: In fact, the next paragraph as well as 3.1.2 Interface properties of PhCs provide explanations as to why the phenols positioned themselves at the air-water interface. Line 202:

"In **Fig. 2(a)**, we observe the three key processes involving PhCs (Ph, 4-HBA, or VL) diffusing into the water slab from the air region. (I) The mutual attraction of gaseous Ph, 4-HBA, or VL and nanoparticles; (II) The uptake of PhCs (Ph, 4-HBA, or VL) at the air-nanoparticle interface; (III) The hydration reaction of PhCs (Ph, 4-HBA, or VL) in the bulk water. **Fig. 2(b)** displays the free energy profile of the trajectories as Ph/4-

HBA/VL transitions from the air into the bulk water (see **Text 6** for calculations details). The $\Delta G_{gas \rightarrow interface}$ values are $-0.22$ kcal mol$^{-1}$ for the Ph-A-W (Phenol-Air-Water) system, $-0.45$ kcal mol$^{-1}$ for the 4-HBA-A-W (4-hydroxybenzaldehyde-Air-Water) system, and $-0.20$ mol$^{-1}$ for the VL-A-W (Vanillin-Air-Water) system. These values suggest that it is thermodynamically favorable for PhCs to approach the interfacial water molecules. **Fig. S3** illustrates typical snapshots from the trajectories of PhCs (Ph, 4-HBA, or VL). Initially, one molecule of Ph, 4-HBA, or VL was placed in the center of the water box, with an equivalent COM distance of 2 nm between the PhCs and the air phase. Subsequently, the PhCs moved closer to the interface, leading to adsorption at the A-W interface. During the adsorption process, the H atom of phenolic hydroxyl group served as an electron donor, binding to the oxygen atom on the surface and preventing its return to the bulk water. Concurrently, hydrogen bonds were formed. This property allowed the phenolic hydroxyl groups on PhCs can effectively adhere to the A-W interface, consistent with the experimental observations using steady-state interfacial vibrational spectra (Kusaka et al., 2021). Based on these findings, the location where air and water meet exhibits an increased concentration of PhCs.

**3.1.2  *Interface properties of PhCs**

The research focused on understanding the behavior of PhCs at A-W interface. The distribution probability of angle ($\alpha$, $\beta$, $\gamma$) for Ph/4-HBA/VL in relation to the A-W interface is shown in **Fig. 3(a)–(d)**. The Z-axis is defined as the axis perpendicular to the interface. The angles are formed between the Z-axis and the benzene ring plane, the phenolic hydroxyl group, and the O–$\alpha$C bound of Ph, 4-HBA, and VL, respectively,

denoted as $\alpha$, $\beta$, and $\gamma$. In the Ph-A-W, 4-HBA-A-W, and VL-A-W systems, a broad distribution range is observed, suggesting that PhCs are rather randomly distributed across the interface. However, statistically, the highest distribution range for $\alpha$ and $\beta$ falls within 15°–25° (or 145°–165°) and 75°–95°, respectively. This applies to both $\alpha$ and $\beta$. In the VL-A-W system, the highest distribution range for $\alpha$ is around 93°. In general, introducing more hydrophilic functional groups increases the characteristic angle $\alpha$ and $\beta$ of PhCs at the interface, allowing for more secure adsorption at the water-air interface.

To set up the interface reaction environment for further quantum chemical calculations, the radial distribution function (g(r)) of Ph, 4-HBA, and VL at interfaces was computed and is show in **Fig. 3 (e)–(g)**. These figures also display the radial distribution function (RDF) and the coordination number N of $H_{Ph-OH}-O_{H_2O}$, $H_{4-HBA-OH}-O_{H_2O}$, and $H_{VL-OH}-O_{H_2O}$ at the A-W interface. Peak intensities are observed in the range of 0.25–0.3 Å for $H_{Ph-OH}-O_{H_2O}$, $H_{4-HBA-OH}-O_{H_2O}$, and $H_{VL-OH}-O_{H_2O}$, as shown in **Fig. 3(e)–(g)**, respectively. The interaction between $H_{PhCs}$ and $O_{H_2O}$ is the primary factor influencing the stability of PhCs at the interface. The N values of $H_{Ph-OH}-O_{H_2O}$, $H_{4-HBA-OH}-O_{H_2O}$, and $H_{VL-OH}-O_{H_2O}$ are 2.68, 2.51, and 2.09 respectively. The number of functional groups attached to the benzene ring affects the N value; more functional groups lead to a lower N value. When a molecule has more functional groups, it occupied more space and exerts a stronger repulsive force on the nearby water molecules compared to those with fewer functional groups".

25) 194-197: The concept of "nanoparticles" appears out of nowhere and is completely wrong.

*Author reply*: Line 204: The "nanoparticles" has been deleted.

26) l. 201- 206: A literature comparison to the free energy change of hydration values is needed.

*Author reply*: A literature comparison has been added to the manuscript. Line 211: "This finding is consistent with previous studies about Per-and poly-fluoroalkyl substances (PFAS) at A-W interface (Yuan et al., 2023) ".

27) l. 211-213: The text is confusing. Explain how an atom of hydrogen serves as an electron donor and state to what electron acceptor. Is the text referring to the hydrogen atom that should participate in hydrogen bonding?

Author reply: This sentence has been modified to the manuscript. Line 220: "During the adsorption process, the H atom of the phenolic hydroxyl group binds to the oxygen atom of the $H_2O$ molecules at the A-W interface, forming H bonds and preventing its return to the bulk water".

28) l. 213: Why should the reader care about the return to the "bulk water"? Shouldn't be more important the return to "air"? The model approached the irrelevant problem and missed the relevant problem questioning its validity.

Author reply: Reactions inside the aqueous particle have also been a hot topic of interest in recent years (Tilgner et al., 2021; Mabato et al., 2023; Zhang et al., 2024; Rana et al., 2024) , so we also focused on the process by which phenolic compounds enter the interior of the droplet.

29) l. 215, l. 302-305, l. 319-325, l. 344-359, l. 367-374, and l. 417-500: Bring here the information from the literature search provided in point 8 above to supplement the discussion with relevant experimental observations of interfacial reactivity and $O_3$ and OH radical initiated reactions.

Author reply: These are updated in this manuscript.

Line 223: "This property allowed the phenolic hydroxyl groups on PhCs can effectively adhere to the A-W interface, consistent with the experimental observations using steady-state interfacial vibrational spectra (Kusaka et al., 2021) and Fourier transform infrared (FTIR) imaging micro-spectroscopy (Guzman et al., 2022) ".

Line 278: "PhCs, once released into the atmosphere, undergo several processes, including adsorption on mineral aerosol surfaces, accumulation at the A-W interface, dispersion in bulk water within liquid droplets, and oxidation reactions initiated by atmospheric oxidants (Lin et al., 2017) ".

Line 294:$O_3$ is a major oxidant in the atmosphere, with high concentrations in the troposphere ranging between $9.85 \times 10^{11}$ molecules cm$^{-3}$ (Tomas et al., 2003; Pillar-Little et al., 2014). Investigating the fate of PhCs in the presence of $O_3$ is essential (Pillar-Little et al., 2014; Rana and Guzman, 2020) ".

Line 298: "The ozonolysis of PhCs involves the synthesis of primary ozonide, the formation of active Criegee intermediate (CI), and the disintegration of CI (Rynjah et al., 2024) ".

Line 323:"The worldwide mean tropospheric concentration of HO$^{\bullet}$ is roughly $11.3 \times 10^5$ molecules cm$^{-3}$ (Lelieveld et al., 2016) ".

Line 426: "These hydroxylation products have been detected by experimental means (Pillar-Little et al., 2014; Pillar-Little and Guzman, 2017; Rana and Guzman, 2020) ".

30) l. 217-218: Explain how you arrive at this conclusive statement for an increased concentration of phenols at the air-water interface and/or moderate its strength.

Author reply: This sentence has been revised. Line 227:

"Based on these findings, compared to the number of PhCs molecules distributed in the gas phase and in bulk water, the location where air and water meet exhibits an increased the number of PhCs molecules".

31) l. 219-249: The section about the interface properties of phenols is not key to the readers of ACP and should be completely moved to the Supplementary Materials document. A clarification of the meaning of "N value" will be needed. The interesting summary statements that should remain in the main manuscript are based on l. 231-234 and l. 242-245. A note should be added to indicate a full description of this part of the work is available in the Supplementary Materials.

Author reply: This section has been revised to lines 232 of the manuscript. "Introducing more hydrophilic functional groups increases the characteristic angle $\alpha$ and $\beta$ of PhCs at the interface, allowing for more secure adsorption at the water-air interface. The interaction between $H_{PhCs}$ and $O_{H_2O}$ is the primary factor influencing the stability of PhCs at the interface. The coordination number (N) of $H_{Ph-OH}-O_{H_2O}$, $H_{4-HBA-OH}-O_{H_2O}$, and $H_{VL-OH}-O_{H_2O}$ are 2.68, 2.51, and 2.09 respectively. The number of functional groups attached to the benzene ring affects the N value; more functional groups lead to a lower N value. The reason is that aldehyde and methoxy are strong electronwithdrawing groups, which will reduce the conjugation effect between the benzene ring and the hydroxyl group, making the hydrogen atom on the hydroxyl group partially positively charged, thus weakening the hydrogen bonding ability with water molecules. See **Text S7** for interface properties of PhCs".

32) l. 247-249: The statement is misleading. What happens if the functional groups are hydrophilic?

Author reply: The statement has been revised. Line 239: "The reason is that aldehyde and methoxy are strong electron-withdrawing groups, which will reduce the conjugation effect between the benzene ring and the hydroxyl group, making the hydrogen atom on the hydroxyl group partially positively charged, thus weakening the hydrogen bonding ability with water molecules".

33) l. 251-257 and l. 268-279: The information is not key to the readers of ACP and should be moved to the Supplementary Materials document. Moreover, the statements in l. 268-279 should be rechecked for technical correction and the interactions of C-O and -OH groups should be re-evaluated.

Author reply: The information has been moved to the Supplementary Materials document. This section of interactions has been revised in **Text S8**.

"In **Fig. S6**, the primary interaction for 4-HBA / VL and $TiO_2$ clusters occurs between the Ti atom and the $O_{-CHO}$ atom, with distances ranging from 1.93 and 2.07 Å".

34) l. 258-267 and l. 281: A close examination of the literature cited in these statements (Bai et al. and Qu and Kroes) shows there is no connection to "phosphorus atoms". This

reviewer finds these statements to compromise the intellectual integrity of the manuscript.

Author reply: The "phosphorus atoms" has been modified to "Ph".

35) l. 286-287: The text appears contradictory as the authors are unaware of the differences between physisorption and chemical adsorption.

Author reply: Sorry, the meaning of this sentence was not clearly expressed and has been revised to line 262: "Physorption energy range from $-1.20$ to $9.56$ kcal mol$^{-1}$ (Nollet et al., 2003), thus this adsorption process in this study is spontaneous chemical adsorption".

36) l. 293, l. 295 and l. 351: The manuscript does not present work with the molecule of "benzene". Scientific accuracy is needed.

Author reply: The "benzene ring" means "benzene ring of PhCs". It has been modified to the manuscript. Line 270:

"Benzene C atom of Ph exhibits sp$^2$ hybridization, meaning it forms one $\sigma$-bond and one $\pi$-bond. The sp$^2$ hybridization of benzene for Ph explains its limited interaction with TiO$_2$ clusters and accounts for the substantial adsorption energy".

Line 300:

"The O$_3$-initiated reactions of Ph/4-HBA/VL involve radical adduct formation (RAF) channels on their benzene ring (R$_{O_3\text{-RAF}}$1–6), highlighted in red in **Fig. S9**. **Fig. 5(a)–(d)** depict that the ozonolysis pathways R$_{O_3\text{-RAF}}$ are exergonic, indicating their spontaneity".

37) l. 309-311 and l. 315: At the air-water interface or in water? Clarification is needed in terms of the significant reduction in molecular energy mentioned.

Author reply: The reduction in the overall energy of the molecule means that the most stable molecular configuration is obtained through optimization. The structural optimization of the PhCs molecules prior to computation was done entirely in the gas phase. This part of the constraint has been added to the manuscript. Line 288: "Moreover, the lone pair electrons of oxygen atoms can form additionally p-$\pi$ conjugations with the $\pi$ electrons of the phenyl ring, further reducing the overall energy of VL in gas phase".

38) l. 318: The idea is this statement is incomplete; provide the full range of relevant $O_3$ molecules and update to more current references.

Author reply: The ozone concentration has been updated in the manuscript.

Line 295: "$O_3$ is a major oxidant in the atmosphere, with high concentrations in the troposphere ranging between $9.85 \times 10^{11}$ molecules cm $^{-3}$ (Tomas et al., 2003; Pillar-Little et al., 2014) ".

39) l. 336-339 and l. 340-343: It appears that based on the current work, the authors are unable to arrive at such a conclusion. How is the comparison for phenols oxidation based and among different media? What is the order of reaction in each media considered?

Author reply: The lower the $\Delta G^{\ddagger}$ values, the more likely an oxidation reaction will take place. However, only the reaction mechanism is considered here, and the detailed analysis of the order of reaction in each media will be continued later in the reaction kinetics section. In **Fig. 4 (e)-(h)**, the $\Delta G^{\ddagger}$ values of $O_3$-initiated reactions with Ph at

the A-W interface are lower than that on other EMs; the $\Delta G^{\ddagger}$ values of O₃-initiated reactions with VL on TiO₂ clusters are lower than that those on other EMs.

[Figure]

**Fig. 4** Statistical charts of calculated **(a) – (d)** $\Delta_r G$ and **(e) – (h)** $\Delta G^{\ddagger}$ values for O₃-initiated reactions; **(i) – (l)** $\Delta_r G$ and **(m) – (p)** $\Delta G^{\ddagger}$ values for HO•-initiated reactions.

In **Fig. 4 (m)-(p)** and **Fig. S9(e)-(h)** and **(m)-(p)**, this is also the case for HO•-initiated reactions.

[Figure]

**Fig. 4** Statistical charts of calculated **(a) – (d)** $\Delta_r G$ and **(e) – (h)** $\Delta G^{\ddagger}$ values for O₃-initiated reactions; **(i) – (l)** $\Delta_r G$ and **(m) – (p)** $\Delta G^{\ddagger}$ values for HO•-initiated reactions.

[Figure]

**Fig. S9** Statistical charts of calculated **(a) – (d)** $\Delta_r G$ and **(e) – (h)** $\Delta G^{\ddagger}$ values for O₃-initiated reactions; **(i) – (l)** $\Delta_r G$ and **(m) – (p)** $\Delta G^{\ddagger}$ values for HO•-initiated reactions.

40) l. 352-355: Based on the current literature from point 8, this is misleading, and the reference used here is irrelevant.

Author reply: Gao et al. (Gao et al., 2019) believed that the reactivity of the single electron transfer mechanism of musk xylene with $HO^\bullet$ is low. In fact, based on our previous calculations (Huo et al., 2024) of the $HO^\bullet$-initiated reactions of neutral aromatic compounds, it was shown that single electron transfer (SET) reactions can be ignored.

41) l. 364: What do the authors mean by the "… $TiO_2$ clusters are the most favorable"?

Author reply: It means that $R_{RAF-HO^\bullet}$ routes are most likely to take place.

42) l. 419: These aromatic compounds do not react with $O_2$. Scientific accuracy is needed.

Author reply: It is not written here that these aromatic compounds do not react with $O_2$.

43) l. 424-426" Why is this a "desirable outcome"? Why is this "most attractive"? Clarification is needed for the readers to understand what the points made are.

Author reply: Because the reaction for the formation of $C_6H_5O$-OO is barrierless, thus it's a desirable outcome. It has been written before this sentence. Line 404:

"As can be seen in **Fig. 7(a),** the addition of $O_2$ to the C3 sites of the $C_6H_5O$ radicals results in the formation of $C_6H_5O$-OO with no barriers in either the gas phase or the bulk water. This is a desirable outcome".

The "most" has been removed to more accurately describe the conclusion. "For the transformation of the $C_6H_5O_2$-OO radicals that were created, the ring closure reaction to form $C_6H_5O_2$-OO-d is attractive option".

44) l. 430-431 and l. 460: This connection came unexpectedly. Improvements are needed to introduce the idea of reaction of "nitric oxide" with the radical. "NO" was not introduced earlier for this system. Why is "NO-O" abstraction a desirable choice?

Author reply: It has been added in line 409:

"In the atmosphere, these Criegee intermediates also may undergo bimolecular reactions with $NO_X$ (Sun et al., 2020) ".

The reason why "NO-O" abstraction is a desirable choice has been added in line 459-455: "On the other hand, in **Fig. S10(a)**, the very low $\Delta G^{\ddagger}$ values (19.74 ~ 22.89 kcal mol$^{-1}$) of the -NO-O abstraction make it a desirable choice".

45) l. 476-482: The manuscript should explain how valid these comparisons are. An expansion to include calculations of reaction rates is needed as constants alone do not communicate much for such systems. What reaction order is considered for each medium? What concentration of each oxidant should be assumed? Literature in point 8 has compared $HO^{\bullet}$ and $O_3$ driven reactions at the air-water interface.

Author reply: This part has been updated in Line 509:

"This study summarizes the $O_3$- and $HO^{\bullet}$-initiated reaction sequences of three PhCs in different environmental media. The reaction sequences for $O_3$- and $HO^{\bullet}$-initiated reactions of Ph and 4-HBA are identical in different environmental media, while VL shows slight variations. For $O_3$-initiated reactions, the reaction sequences are as follows: Ph: A-W interface > Bulk water > Gas phase > $TiO_2$ clusters; 4-HBA: Bulk water > A-W interface > $TiO_2$ clusters > Gas phase; VL: Bulk water > $TiO_2$ clusters > A-W interface > Gas phase. For $HO^{\bullet}$-initiated reactions, the sequences are: Ph: A-W interface

≈ Bulk water > Gas phase > TiO$_2$ clusters; 4-HBA: Bulk water > A-W interface > TiO$_2$ clusters > Gas phase; VL: TiO$_2$ clusters > Bulk water > A-W interface > Gas phase. See **Text S9** for ecotoxicity assessment. According to the atmospheric concentration of O$_3$, the atmospheric lifetime of Ph is the shortest (< 1s) of the three PhCs, whereas 4-HBA and VL were oxidized more slowly than Ph (Smith et al., 2016)".

46) Fig. 2: Specify if you are dealing with the deltaG of hydration or what in the y-axes. The meaning of nanoparticles appears inappropriate for this field.

Author reply: The meaning of "deltaG" has been written in the photo captions:

"**(b)** free energy change profile of gaseous PhCs (Ph, 4-HBA, or VL) approaching the bulk water". Besides, the "nanoparticles" has been modified to " water drops ".

47) Fig. 3: The quality of this image is not the optimum expected for publication in ACP. Moreover, after improving it, this figure should be moved to the Supplementary Materials document.

Author reply: The resolution of **Fig. 3** has been increased. The **Fig.3** has been moved to Supplementary Materials document.

48) Fig. 7: The resolution is poor for reading it. How are both "gas phase" and "bulk water" included together in panel (a)? How does the bottom reaction of panel (a) loose OH? It seems unlikely based on extensive studies in point 8. The final product should have one more OH.

Author reply: The resolution of **Fig. 6** has been increased. A different calculation method was used. The photo captions have been written. "**Fig.6** Subsequent reaction mechanisms of important intermediates (IMs) (unit: kcal mol$^{-1}$) in **(a)** gas phase (g) /

bulk water (l) and at **(b)** A-W interface".

The bottom reaction of panel (a) is that the transfer of hydrogen atoms accompanied by the removal of HO$_2$ radicals.

The Final product has been updated.

**Fig.6** (Continue) Subsequent reaction mechanisms of important intermediates (IMs) (unit: kcal mol$^{-1}$) in **(a)** gas phase (g) / bulk water (l) and at **(b)** A-W interface.

How is the second reaction in the bottom of panel (b) capable of losing OH? The no go reaction (red cross) contradicts thermodynamic measurements that have been experimentally determined and are available in the literature of point 8. The material in the figure needs significant improvements as well as its discussion based on published literature that has not been included in the manuscript.

Similar comments from point 48 are valid also for Fig. S10.

Author reply: The second reaction in the bottom of panel (b) loses the $HO_2^\bullet$, not the $HO^\bullet$. It takes the H atom from the $HO^\bullet$ addition site next to it via O−O abstraction while the $HO_2^\bullet$ is formed. However, this direct formation of the $HO_2^\bullet$ is more difficult to form later in the reaction, so this later step is carried out in stages. Reactions marked with a cross cannot be generated directly by electron transfer reactions, but may be generated by other pathways.

Yes, but one of the reaction conditions in this study was in the troposphere in the presence of mineral aerosols. Thus, reactions in the material domain were not investigated in this study.

Literature research (Rana and Guzman, 2022c) revealed that the hydrogen on the benzene ring of the reactant in S11 can eventually be replaced by a hydroxyl group.

49) Fig. 8: What phase/medium is this figure referring to? Many experimental values for such constants have been reported in the literature based on experimental work missing from this manuscript.

Author reply: "**Fig.8** Calculated rate constants for the initial reactions of Ph, 4-HBA, and VL with $O_3$ and $HO^\bullet$ at different temperatures (278 – 318 K) and 1 atm at A-W interface and on the $TiO_2$ clusters".

[revised manuscript text omitted]

Tilgner, A., Schaefer, T., Alexander, B., Barth, M., Collett Jr, J. L., Fahey, K. M., Nenes, A., Pye, H. O. T., Herrmann, H., and McNeill, V. F.: Acidity and the multiphase

chemistry of atmospheric aqueous particles and clouds, Atmos. Chem. Phys., 21, 13483-13536, https://doi.org/10.5194/acp-21-13483-2021, 2021.

Tomas, A., Olariu, R. I., Barnes, I., and Becker, K. H. J. I. J. o. C. K.: Kinetics of the reaction of O3 with selected benzenediols, 35, 223-230, https://doi.org/10.1002/kin.10121, 2003.

Vione, D., Maurino, V., Minero, C., Duncianu, M., Olariu, R.-I., Arsene, C., Sarakha, M., and Mailhot, G.: Assessing the transformation kinetics of 2- and 4-nitrophenol in the atmospheric aqueous phase. Implications for the distribution of both nitroisomers in the atmosphere, Atmos Environ, 43, 2321-2327, https://doi.org/10.1016/j.atmosenv.2009.01.025, 2009.

Wang, R., Li, K., Li, J., Tsona, N. T., Wang, W., and Du, L.: Interaction of Acrylic Acid and SO2 on the Surface of Mineral Dust Aerosol, Acs Earth Space Chem, 7, 548-558, https://doi.org/10.1021/acsearthspacechem.2c00323, 2023.

Xu, C. and Wang, L.: Atmospheric Oxidation Mechanism of Phenol Initiated by OH Radical, J Phys Chem A, 117, 2358-2364, https://doi.org/10.1021/jp308856b, 2013.

Yuan, S., Wang, X., Jiang, Z., Zhang, H., and Yuan, S.: Contribution of air-water interface in removing PFAS from drinking water: Adsorption, stability, interaction and machine learning studies, Water Research, 236, 119947, https://doi.org/10.1016/j.watres.2023.119947, 2023.

Zein, A. E., Coeur, C., Obeid, E., Lauraguais, A., and Fagniez, T.: Reaction Kinetics of Catechol (1,2-Benzenediol) and Guaiacol (2-Methoxyphenol) with Ozone, J Phys Chem A, 119, 6759-6765, https://doi.org/10.1021/acs.jpca.5b00174, 2015.

Zhang, J., Shrivastava, M., Ma, L., Jiang, W., Anastasio, C., Zhang, Q., and Zelenyuk, A.: Modeling Novel Aqueous Particle and Cloud Chemistry Processes of Biomass Burning Phenols and Their Potential to Form Secondary Organic Aerosols, Environ. Sci. Technol., 58, 3776-3786, https://doi.org/10.1021/acs.est.3c07762, 2024.

Zhang, W., Ji, Y., Li, G., Shi, Q., and An, T.: The heterogeneous reaction of dimethylamine/ammonia with sulfuric acid to promote the growth of atmospheric nanoparticles, Environ Sci-Nano, 6, 2767-2776, https://doi.org/10.1039/C9EN00619B, 2019.

Zhang, W., Wang, X., Pan, H., Zeng, X., Li, G., Liu, H., Kong, J., Zhao, H., and An, T.: Experimental and DFT investigations on adsorption–regeneration performance and deactivation mechanism over engineered carbon fiber: role of pore structure and functional groups, Environ Sci-Nano, 10, 2790-2798, https://doi.org/10.1039/D3EN00443K, 2023.

Zhong, J., Kumar, M., Anglada, J. M., Martins-Costa, M. T. C., Ruiz-Lopez, M. F., Zeng, X. C., and Francisco, J. S.: Atmospheric Spectroscopy and Photochemistry at Environmental Water Interfaces, Annu Rev Phys Chem, 70, 45-69, https://doi.org/10.1146/annurev-physchem-042018-052311, 2019.

---

## Author Response (AR2)

**MS No.:** egusphere-2023-2856

**TITLE:** *Rapid oxidation of phenolic compounds by O₃ and HO•: effects of air-water interface and mineral dust in tropospheric chemical processes*

Dear editor,

Thank you for your comments on our manuscript to enrich the content of this work. The first author of our manuscript, Yanru Huo, has recently started a postdoctoral position at McGill University. As a result, we would like to update the author's affiliation to include McGill University as the second affiliation. Due to the change in the type of our manuscript to a Research Article, we have made a small adjustment to the content. Specifically, the "Data Availability" section has been removed as all relevant data are now included in the supporting materials. We have tried our best to modify the manuscript according to your suggestions. Each question was answered in blue; every correction is in red, and the deleted texts are marked with a delete line number in this response. Besides, every correction is in red in the revised manuscript. The detailed response to each comment is shown as follows:

**Reviewer #1:**

Question 1: Please provide a better explanation for choosing phenol, 4-hydroxybenzaldehyde and vanillin. For example, are these species observed in the atmosphere, and are they major types of phenolic compounds in the atmosphere?

*Author reply*: The manuscript has been modified in Line 106, "Increasing the number of constituents on the aromatic ring would affect the reactivity and lead to complex compounds after reaction addition and/or open ring pathways. Phenol (Ph), 4-hydroxybenzaldehyde (4-

HBA), and vanillin (VL) are typical lignin pyrolysis products (Jiang et al., 2010; Kibet et al., 2012) ".

Question 3: please modify the manuscript to address this comment. Answering the question in just the "response to referees" is not sufficient.

*Author reply*: The manuscript has been modified:

Line 175 "After analyzing the stability of the wavefunction, the method we used is reliable".

Line 178 "The frequency correction factor (0.967) has been taken into account".

**Reviewer #2:**

Question 5: The authors state that "... the troposphere ranging between 9.85*10^11 molec cm^-3". Please indicate a lower limit. If you say a range, there needs to be an upper and lower limit.

*Author reply*: The ozone concentration has been updated in the manuscript in Line 304:

"$O_3$ is a major oxidant in the atmosphere, with high concentrations in the troposphere of $9.85 \times 10^{11}$ molecules cm$^{-3}$ (Tomas et al., 2003; Pillar-Little et al., 2014) ".

**Reviewer #3:**

Question 13, 14, 19, 28, 41: Please modify the manuscript to address these comments/questions. Answering the questions in just the "response to referees" is not sufficient.

Question 13) l. 130-131: The manuscript should state how the random selection was executed in this work.

*Author reply*: The manuscript has been revised in line 139:

"Prior to the formal simulation, six Ph molecules were randomly selected position placed in a vacuum above the water box for 150 nanoseconds of NVT molecular dynamics simulations".

Question 14: l. 132: The manuscript should explain to the readers of ACP why such a short 150 ns period is relevant. The explanation should be followed by establishing the relevance of this period to what happens in the environment.

*Author reply*: It appears that there might be some confusion between the timescale used in molecular dynamics simulations and the real-world timescale of environmental processes. Molecular simulations, though conducted over short periods like 150 ns, are designed to capture essential molecular dynamics that occur on this time scale, such as bond formation, conformational changes, and interaction events.

These simulations provide detailed insight into microscopic mechanisms, which, although happening in a simulated timeframe, can offer valuable predictive power about long-term macroscopic processes. The results from such simulations can be extrapolated to understand how molecules behave under real-world environmental conditions. Thus, the 150 ns simulation period is highly informative for understanding key molecular interactions that are relevant on a larger environmental timescale.

Line 141: In order to make the experimental scientists understand that the time of the molecular simulation and the time when the reaction occurs are not one and the same concept, clarification was made in the manuscript. "The purpose of simulating 150 ns is to capture the fundamental molecular dynamics that occur on this time scale, such as bond formation, conformational changes, and interaction events".

Question 19: l. 159: What is the meaning of "benchmarking" here? Clarify the text.

*Author reply*: It has been modified in the manuscript. Line 172: "Calculated at the CCSD(T)/cc-pVDZ, CBS-QB3, B3LYP/6-311+G(d,p), MP2/6-311+G(d,p) and M06-2X/6-311+G(d,p) levels, Cao et al. (Cao et al., 2021) found that M06-2X/6-311++G(3df,2p)//M06-2X/6-31+G(d,p) is reliable for PhCs at gas phase".

Question 28: l. 213: Why should the reader care about the return to the "bulk water"? Shouldn't be more important the return to "air"? The model approached the irrelevant problem and missed the relevant problem questioning its validity.

*Author reply:* This part of the response has been added to the manuscript in Line 289:

"Reactions inside the aqueous particle have also been a hot topic of interest in recent years (Tilgner et al., 2021; Mabato et al., 2023; Zhang et al., 2024; Rana et al., 2024) , so we also focused on the process by which phenolic compounds enter the interior of the droplet".

Question 41: l. 364: What do the authors mean by the "… $TiO_2$ clusters are the most favorable"?

*Author reply:* The manuscript has been revised in line 351:

"Among the three aromatic compounds, the $R_{RAF-HO^\bullet}$ routes of VL on $TiO_2$ clusters are most likely to take place (**Fig. 4(n)**) ".

Question 35: should "physorption" be changed to "physisorption".

*Author reply*: In Line 269, "physorption" has been changed to "physisorption".

Question 42: I think the referee is indicating that you should not state that "aromatic compounds react with $O_2$". The mechanism is more complicated than a direct reaction, so your statement is not exactly accurate.

*Author reply*: The manuscript has been revised in line 406:

"For the purposes of this discussion, the primary atmospheric fate of the selected aromatics was considered to be their reactions with $O_2$ (typically mediated by reactive intermediates or catalytic processes) and $O_3$".